# The Atr-Chek1 pathway inhibits axon regeneration in response to Piezo-dependent mechanosensation

Feng Li[1,2,10], Tsz Y. Lo[1,10], Leann Miles [3,10], Qin Wang [1,2,10], Harun N. Noristani [4,5], Dan Li[1], Jingwen Niu[4], Shannon Trombley[1], Jessica I. Goldshteyn[1], Chuxi Wang[1], Shuchao Wang[1], Jingyun Qiu[1], Katarzyna Pogoda [6,7], Kalpana Mandal[6], Megan Brewster[8], Panteleimon Rompolas[8], Ye He[9], Paul A. Janmey[6], Gareth M. Thomas[4,5], Shuxin Li[4,5] & Yuanquan Song [1,2✉]

Atr is a serine/threonine kinase, known to sense single-stranded DNA breaks and activate the DNA damage checkpoint by phosphorylating Chek1, which inhibits Cdc25, causing cell cycle arrest. This pathway has not been implicated in neuroregeneration. We show that in *Drosophila* sensory neurons removing Atr or Chek1, or overexpressing Cdc25 promotes regeneration, whereas Atr or Chek1 overexpression, or Cdc25 knockdown impedes regeneration. Inhibiting the Atr-associated checkpoint complex in neurons promotes regeneration and improves synapse/behavioral recovery after CNS injury. Independent of DNA damage, Atr responds to the mechanical stimulus elicited during regeneration, via the mechanosensitive ion channel Piezo and its downstream NO signaling. Sensory neuron-specific knockout of Atr in adult mice, or pharmacological inhibition of Atr-Chek1 in mammalian neurons in vitro and in flies in vivo enhances regeneration. Our findings reveal the Piezo-Atr-Chek1-Cdc25 axis as an evolutionarily conserved inhibitory mechanism for regeneration, and identify potential therapeutic targets for treating nervous system trauma.

[1] Raymond G. Perelman Center for Cellular and Molecular Therapeutics, The Children's Hospital of Philadelphia, Philadelphia, PA, USA. [2] Department of Pathology and Laboratory Medicine, University of Pennsylvania, Philadelphia, PA, USA. [3] The Graduate Group in Biochemistry and Molecular Biophysics, University of Pennsylvania, Philadelphia, PA, USA. [4] Shriners Hospitals Pediatric Research Center (Center for Neurorehabilitation and Neural Repair), Temple University School of Medicine, Philadelphia, PA, USA. [5] Department of Anatomy and Cell Biology, Temple University School of Medicine, Philadelphia, PA, USA. [6] Institute for Medicine and Engineering, University of Pennsylvania, Philadelphia, PA, USA. [7] Institute of Nuclear Physics, Polish Academy of Sciences, Krakow, Poland. [8] Department of Dermatology, University of Pennsylvania, Philadelphia, PA, USA. [9] The City University of New York, Graduate Center - Advanced Science Research Center, Neuroscience Initiative, New York, NY, USA. [10] These authors contributed equally: Feng Li, Tsz Y. Lo, Leann Miles, Qin Wang. ✉email: songy2@email.chop.edu

Axon regeneration in the adult central nervous system (CNS) is rather limited, due to the diminished regenerative potential of mature neurons and the inhibitory microenvironment[1–4]. As a result, permanent disability often occurs in individuals with spinal cord injury or stroke. In contrast, neurons in the peripheral nervous system (PNS) are generally capable of robust axon regeneration and regain functional recovery after injury. However, in cases of severe peripheral nerve insults such as proximal nerve lesions or complete transections, neurological deficits can still occur due to the slow rate of spontaneous axon regeneration, failure of reinnervation, or the development of chronic pain[5]. In humans particularly, minimal clinically meaningful restoration of motor function has been observed[6–8]. Moreover, the regenerative capacity of the PNS declines with aging[9]. Therefore, strategies need to be developed to increase the rate and/or extent of axon regeneration to improve functional outcomes in the adult mammalian CNS and PNS.

To achieve this goal, a major focus of research is to identify the neuronal intrinsic molecular machinery that triggers the regenerative response or acts as a regeneration brake. Various injury paradigms in different model organisms have been established[10] that serve as screening platforms. We previously developed a *Drosophila* peripheral sensory neuron injury model based on laser axotomy. We showed that fly dendritic arborization (da) sensory neurons display regeneration cell-type specificity: class IV but not class III da neurons are able to regenerate[11]. Utilizing this tool, we have performed genetic screens and identified the RNA processing enzyme Rtca as an evolutionarily conserved inhibitor of axon regeneration, which links axon injury to ER stress and RNA modifications[12]. We have since performed an additional screen on other cellular stress pathways, focusing on the DNA-damage response (DDR). We found that mediators of the DNA single-strand break (SSB) response specifically inhibit axon regeneration.

SSBs are known to activate Atr (ataxia telangiectasia and Rad3 related), a serine/threonine kinase that directly phosphorylates Chek1 (checkpoint kinase 1). Chek1 in turn phosphorylates and inhibits the phosphatase Cdc25C (cell division cycle 25C) or Cdc25A, which would prevent Cdk1 (cyclin-dependent kinase 1)/ CycB (cyclin B) from being dephosphorylated and therefore cause a cell cycle arrest in G2/M or S-phase, respectively[13,14]. A multistep model has been proposed for Atr checkpoint activation in response to DNA damage[15], which involves DNA-damage sensing, signal transduction, and execution. DNA damage generates ssDNA (single-stranded DNA), which is recognized and coated by RPA (replication protein A). The primed ssDNA recruits Atr-Atrip (Atr interacting protein) and facilitates the loading of 9-1-1 (Rad9-Hus1-Rad1) by the Rad17 complex. The 9-1-1 complex may then stimulate the kinase activity of Atr-Atrip, leading to phosphorylation of its substrates including Rad17 and Rad9. Phosphorylated Rad17 and Rad9 may facilitate the recruitment of downstream signaling proteins Claspin and TopBP1 (topoisomerase (DNA) II binding protein 1), allowing them to be efficiently phosphorylated by Atr. Phosphorylated TopBP1 may further stimulate the kinase activity of Atr, whereas phosphorylation of Claspin may promote the phosphorylation and activation of Chek1.

Atr can also be activated by mechanical force. It has been reported that Atr can respond to mechanical stimuli, such as osmotic stress, in mediating chromosome dynamics, which is independent of DNA damage[16]. However, the underlying mechanoreceptor remains unknown. We have recently demonstrated that the mechanosensitive (MS) ion channel Piezo is activated during axon regrowth, leading to local elevation of calcium transients and the activation of the Nos (nitric oxide synthase) cascade to restrict axon regrowth, and that Piezo loss of function (LOF) promotes class III da neuron axon regeneration[17]. The downstream cellular and molecular signaling of Piezo-Nos, however, remains elusive. Moreover, mammalian Piezo1 can be activated by osmotic stress[18] and also functions as a regeneration inhibitor[17]. Here, we show that Atr-Chek1 and the associated checkpoint complex act downstream of Piezo to suppress axon regeneration by inactivating Cdc25-Cdk1. Independent of the canonical ssDNA-RPA sensing process, Atr responds to the mechanical stress elicited after axon injury, with Piezo as the mechanosensor and NO (nitric oxide) as the mediator. We further show that blocking Atr-Chek1 promotes axon regeneration both in the PNS and CNS, leading to synapse regeneration and behavioral recovery. The function of Atr-Chek1 in inhibiting axon regeneration appears to be evolutionarily conserved in mammals. This study identifies an unexpected role of the Atr-Chek1 kinase cascade in regulating neuroregeneration, reveals a mechanistic link to the mechanosensitive ion channel Piezo, and provides potential therapeutic targets for stimulating nerve repair.

## Results

**Atr-Chek1-Cdc25-Cdk1 regulate axon regeneration**. We used the previously described *Drosophila* da sensory neuron injury model[11,12] to study axon regeneration. In brief, using a two-photon laser, we injured the axon of the mechanosensitive class III da neurons (labeled with *19-12-Gal4 > CD4tdGFP, repo-Gal80*) in the PNS of early 3rd instar larvae. Degeneration of the distal axon was confirmed at around 24 h after injury (h AI) and regeneration was assessed at around 72 h AI (Fig. 1a). In contrast to wild-type (WT) class III da neurons which failed to regenerate (Fig. 1a, arrow), in a null mutant[19] of the *Drosophila* homolog of Atr, *meiotic 41/mei41²⁹ᴰ*, new axons regrew substantially beyond the injury site (Fig. 1a, arrowheads). The function of Atr/mei41 is cell-autonomous because its RNAi knockdown in class III da neurons (*19-12-Gal4 > mei41 RNAiv103624*) recapitulated the enhancement of regeneration. As expected, class III da neuron-specific RNAi knockdown of the fly homolog of Chek1—grapes/ grp (*19-12-Gal4 > grp RNAiBL27277* and *19-12-Gal4 > grp RNAiv10076*), or *grpᴬ¹⁹⁶* mutant clones of class III da neurons (using MARCM[20]) also enhanced axon regeneration, similarly to Atr/mei41 deficiency (Fig. 1a, arrowheads). On the other hand, class III da neuron-specific overexpression of twine/twe or string/ stg (*19-12-Gal4 > twe* or *19-12-Gal4 > stg*), the fly homologs of Cdc25C/Cdc25A which are negatively regulated by Chek1, promoted axon regeneration (Fig. 1a, arrowheads). Cdc25 is known to activate Cdk1 by removing the inhibitory phosphorylation at tyrosine 15 (Y15) and the adjacent threonine (T14) residues[14]. We therefore overexpressed the phospho-acceptor mutant of Cdk1, Cdk1.T14A.Y15F, which renders it unable to be phosphorylated and is therefore the activated form[21], in class III da neurons (*19-12-Gal4 > Cdk1.T14A.Y15F*). We found that it was sufficient to trigger axon regeneration (Fig. 1a, arrowheads). Moreover, knocking down Cdc25/twe in class III da neurons in the *mei41²⁹ᴰ* background abolished the enhanced regeneration phenotype (Fig. 1a, arrow), consistent with a model in which Cdc25/twe lies downstream of Atr/mei41 to regulate axon regeneration. The regeneration phenotype was further quantified by assessing the "Regeneration percentage" and "Regeneration index" (Fig. 1b, c, Supplementary Fig. 1a, b and "Methods"), as described previously[11]. We also quantified class III da neuron dendrite branching after blocking the Atr pathway and did not observe obvious change in total dendrite length (Supplementary Fig. 1c), suggesting a specific role of this pathway in axon regeneration.

Conversely, we determined whether gain of function (GOF) of Atr-Chek1 would reduce the regenerative potential of class IV da

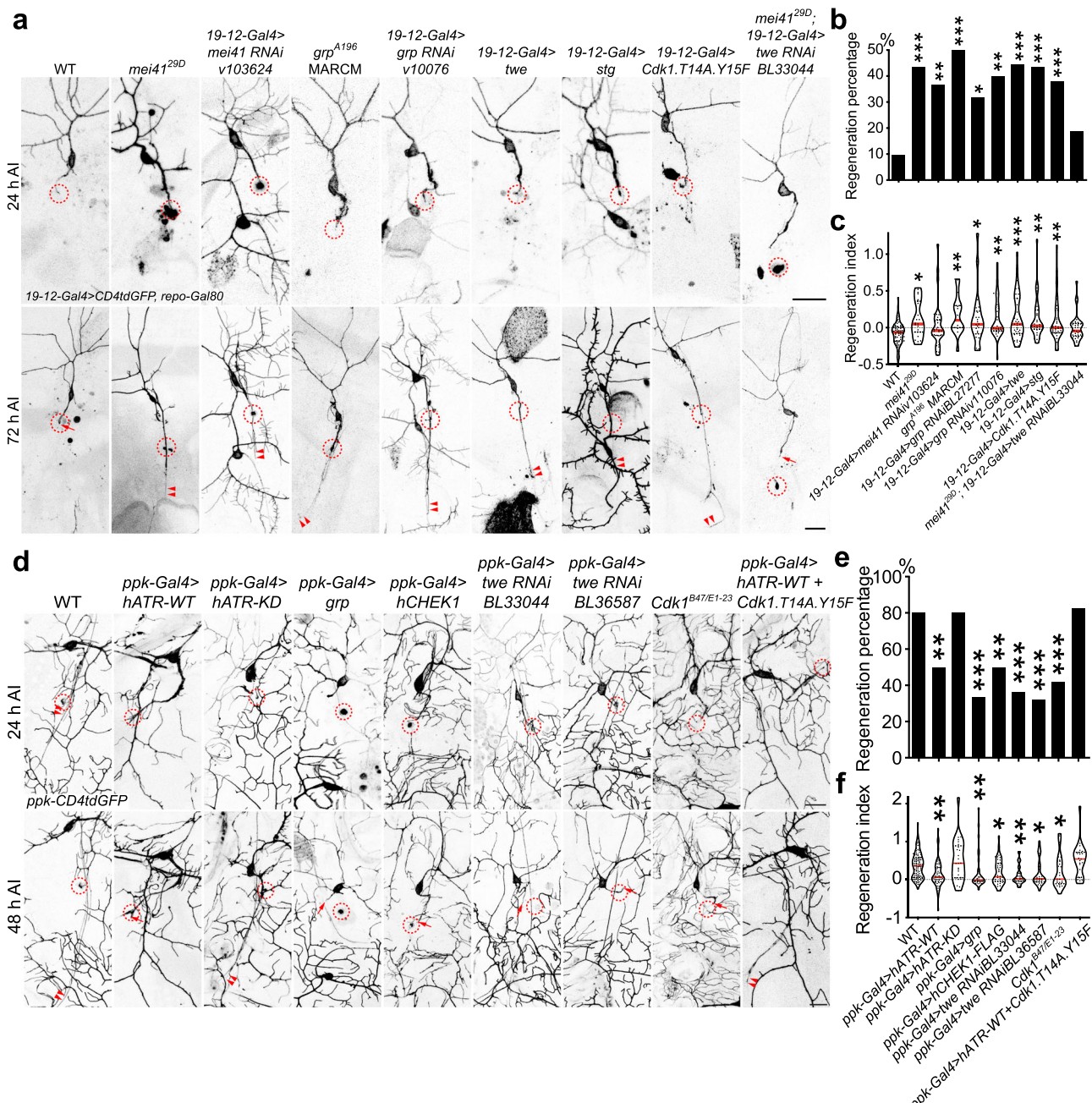

**Fig. 1 The Atr/mei41-Chek1/grp pathway regulates axon regeneration in da sensory neurons in flies. a** Class III da neuron axons fail to regenerate in WT. Atr/mei41 removal as in *mei41^29D* mutants or class III da neuron-specific RNAi leads to increased axon regeneration. *Chek1/grp^A196* mutant clones (with MARCM), class III da neuron expression of Chek1/grp RNAis, Cdc25/twe/stg, or the dephosphorylated/activated Cdk1.T14A.Y15F increases axon regeneration. Class III da neuron expression of twe RNAi suppressed the enhanced regeneration in *mei41^29D* mutants. The injury site is demarcated by the dashed circle. Arrow marks axon stalling while arrowheads show the regrowing axon tips. **b, c** Quantifications of class III da neuron axon regeneration with regeneration percentage (**b**) and regeneration index (**c**). N = 72, 23, 30, 16, 22, 30, 36, 30, 37, and 28 neurons from 6 to 20 larvae. P = 0.0007, 0.003, 0.0007, 0.0177, 0.0013, <0.0001, 0.0002, 0.0008, 0.1818. **d** Class IV da neurons robustly regenerate in WT. Class IV da neuron-specific expression of hATR-WT, grp, hCHEK1, twe RNAis, or LOF of Cdk1 as in transheterozygotes of *Cdk1^B47/E1-23* impedes axon regeneration, whereas the kinase-dead (KD) mutant of hATR fails to show a significant effect. Overexpression of hATR-WT together with the constitutively active Cdk1 (T14A, Y15F) fails to inhibit axon regeneration. **e, f** Quantifications of class IV da neuron axon regeneration. N = 97, 50, 25, 30, 38, 33, 28, 24, and 23 neurons from 6 to 18 larvae. P = 0.0002, >0.9999, <0.0001, 0.0007, <0.0001, <0.0001, 0.0005, >0.9999. *P < 0.05, **P < 0.01, ***P < 0.001 by two-sided Fisher's exact test (**b** and **e**), one-way ANOVA followed by Holm–Sidak's test (**c**) or Dunn's test (**f**). Scale bar = 20 μm. Source data are provided as a Source data file.

neurons, which are normally capable of regeneration[11]. We labeled class IV da neurons with *ppk-CD4tdGFP* and used the following injury paradigm: axotomy was induced at 3rd instar, degeneration was confirmed at 24 h AI and regeneration was assayed at 48 h AI[12]. Compared to WT class IV da neurons,

which exhibited axon regeneration ~80% of the time (Fig. 1d–f, arrowheads), overexpression of wild-type human ATR (hATR-WT) in class IV da neurons significantly reduced the axon regeneration percentage to 50% and decreased the length of the regrown axons (Fig. 1d–f, arrows). However, its kinase-dead

version (hATR-KD) did not significantly alter regeneration (Fig. 1d–f, arrowheads), indicating that the kinase activity of Atr is required for regeneration inhibition. Consistent with this finding, overexpression of Chek1/grp or human CHEK1 (hCHEK1) also led to reduced class IV da neuron axon regeneration (Fig. 1d–f, arrows). The fact that human ATR and CHEK1 are both capable of inhibiting axon regeneration in flies suggests that the role of Atr-Chek1 in mediating regeneration is evolutionarily conserved. This was further confirmed by analyzing their role in mammalian axon regeneration (see below). Moreover, RNAi knockdown of Cdc25/twe in class IV da neurons, or LOF of Cdk1[22,23] as in transheterozygotes of $Cdk1^{B47/E1-23}$ impeded axon regeneration to a similar extent (Fig. 1d–f, arrows). Last, hATR-WT failed to inhibit axon regeneration when co-expressed with the constitutively active Cdk1 (T14A, Y15F) (Fig. 1d–f), confirming that Cdk1 functions downstream of Atr in regulating axon regeneration. Together, these LOF and GOF analyses demonstrate that the Atr-Chek1 cascade modulates neuroregeneration, with Atr/mei41-Chek1/grp and Cdc25/twe-Cdk1 functioning as anti- and pro-regeneration factors, respectively.

We next examined the expression pattern of Atr/mei41 using a transgenic fly that contains a fosmid clone of the FLAG-tagged mei41 genomic locus, mei41::FLAG, so that FLAG expression reflects the endogenous pattern of mei41 at the physiological level[24]. We found that mei41::FLAG is present in class III da neurons, restricted within the nucleus (Supplementary Fig. 1d, red dashed circle). We did not detect obvious difference in the expression level or distribution of mei41::FLAG between uninjured and injured class III da neurons at 24 or 48 h AI (Supplementary Fig. 1d). The expression of mei41::FLAG was also found in other types of da neurons, including class IV da neurons (Supplementary Fig. 1d, yellow dashed circle), suggesting that Atr/mei41 per se is likely not a determining factor for the regeneration cell-type specificity.

To determine the specificity of the Atr-Chek1 pathway in regulating regeneration, we queried the other classical DDR branch which is triggered by double-strand DNA breaks (DSB). Once DSBs are generated, Atm (ataxia telangiectasia mutated) is recruited by the Mre11-Rad50-Nbs1 (MRN) complex to sites of the broken DNA and phosphorylates downstream substrates such as Chek2 (checkpoint kinase 2)[13]. The fly homologs of Atm, Rad50, and Nbs are telomere fusion/tefu, rad50, and nbs1, respectively. We found that their LOF mutations, as in $tefu^{atm-6}$ [25], $rad50^{EP1}$ [26], and $nbs1$ [27] did not result in significant defects in class IV da neuron axon regeneration (Supplementary Fig. 2a, b). Furthermore, LOF of Atm/tefu as in $tefu^{atm-6/atm-3}$ [25] did not lead to enhanced axon regeneration in class III da neurons (Supplementary Fig. 2c–e, arrow). These observations highlight the unique role of the Atr-Chek1 pathway in mediating neuroregeneration and also raise the question whether DNA damage is indeed involved. Moreover, we examined the axon regeneration phenotype in the regeneration-incompetent class I da neurons[11] and found that Atr/mei41 mutants also showed increased regeneration (Supplementary Fig. 2f–h, arrowheads). This suggests that the regeneration–inhibition function of the Atr pathway is applicable to multiple neuronal cell types, as is further exemplified by our regeneration studies in mammals (see below).

**Atr-Chek1 inhibits axon regeneration likely independent of DNA damage.** In the DDR, ssDNAs induced by DNA damage are sensed by RPAs, which then recruit and activate Atr, orchestrated by additional factors. The imminent question is whether DNA damage is implicated in Atr activation during neuroregeneration. To address this question, we first assessed whether

DNA damage is induced after axon injury in class III da neurons. γH2AX (H2A histone family, member X), the serine 139 phosphorylated form of H2AX, serves as the gold standard DNA-damage marker[28,29]. Its fly homolog is His2Av and a phospho-specific antibody against His2Av—p-His2Av has been widely used for detecting DNA damage in flies[30]. Using this antibody, we found there was no difference, in terms of p-His2Av staining, between injured and uninjured class III da neurons at 5 min, 24 h, or 48 h AI (Fig. 2a, b, dashed circles), suggesting that DNA damage is not significantly induced after axon injury or during axon regeneration in da sensory neurons. As a positive control for the antibody, we confirmed that p-His2Av staining is significantly increased in the larval brain after ultraviolet (UV) irradiation (Supplementary Fig. 3a-c). Second, we tested whether His2Av LOF affects axon regeneration and found that class III da neuron-specific knockdown of His2Av with a previously reported RNAi[31] did not increase axon regeneration (Fig. 2c–e, arrow). Third, we asked whether blocking the ssDNA sensing step by eliminating RPAs would interfere with Atr's function in regeneration. RPA is a heterotrimer composed of three subunits Rpa1, Rpa2, and Rpa3. Therefore, we expressed in class III da neurons RNAis targeting their fly homologs RpA-70, RPA2, and RPA3 and found no enhancement of axon regeneration (Fig. 2c–e, arrow). This was further confirmed by the $RPA2^{KG00759}$ amorphic mutant[32] class III da neurons (using MARCM) and the $RPA3^{G0241}$ LOF allele[33] (Fig. 2c–e, arrow). Last, overexpression of RpA-70, RPA2, or RPA3 in class IV da neurons did not reduce their axon regeneration (Supplementary Fig. 3d, e). These data suggest that the neuronal intrinsic DDR may not contribute significantly to injury-induced axon regeneration in da sensory neurons, and that the Atr-Chek1 pathway inhibits axon regeneration independent of its canonical role in DNA damage.

**The Atr-associated checkpoint complex inhibits axon regeneration.** The RPA-mediated DNA-damage sensing step is thus dispensable for Atr's inhibition of axon regeneration. However, we wondered whether other factors in the checkpoint complex, which are important for the signal transduction and execution steps, facilitate the regulation of regeneration. We focused on Atrip, Rad17, the 9-1-1 complex composed of Rad9-Hus1-Rad1, TopBP1, and Claspin (Fig. 3b). We found that class III da neuron-specific RNAi knockdown targeting their fly homologs Atrip/mus304, Rad17, Rad1, TopBP1/mus101, and Claspin all markedly increased axon regeneration (Fig. 3a, c, d, arrowheads). The regeneration enhancement phenotype via TopBP1/mus101 RNAi knockdown was recapitulated in a LOF mutation, $mus101^A$ (Fig. 3a, c, d, arrowheads). Moreover, LOF of Hus1-like, the fly homolog of Hus1, via an insertional allele $Hus1-like^{MI11259}$ which abolished its expression (Supplementary Fig. 4b, c), also promoted axon regeneration (Fig. 3a, c, d, arrowheads). Because the Hus1-like insertion was also mapped to the promoter region of a neighboring gene, ctrip (Supplementary Fig. 4a), we thus analyzed an insertional mutant of ctrip—$ctrip^{MI14762}$, but did not observe a regeneration phenotype in class III da neurons (Supplementary Fig. 4d). These data indicate that the checkpoint complex known to be required for transducing the DNA-damage signal is also essential for facilitating Atr-Chek1 to cell-autonomously inhibit neuroregeneration.

In order to further assess the involvement of the Atr-Chek1 pathway members in sensory neuron regeneration, we determined the expression pattern of the relevant molecules we analyzed in Drosophila and mammals. First, we examined the expression of Cdc25/twe in da neurons with antibody[34] staining and found it was present in both uninjured and injured class III da neurons, similar to Atr/mei41 (Supplementary Fig. 5a).

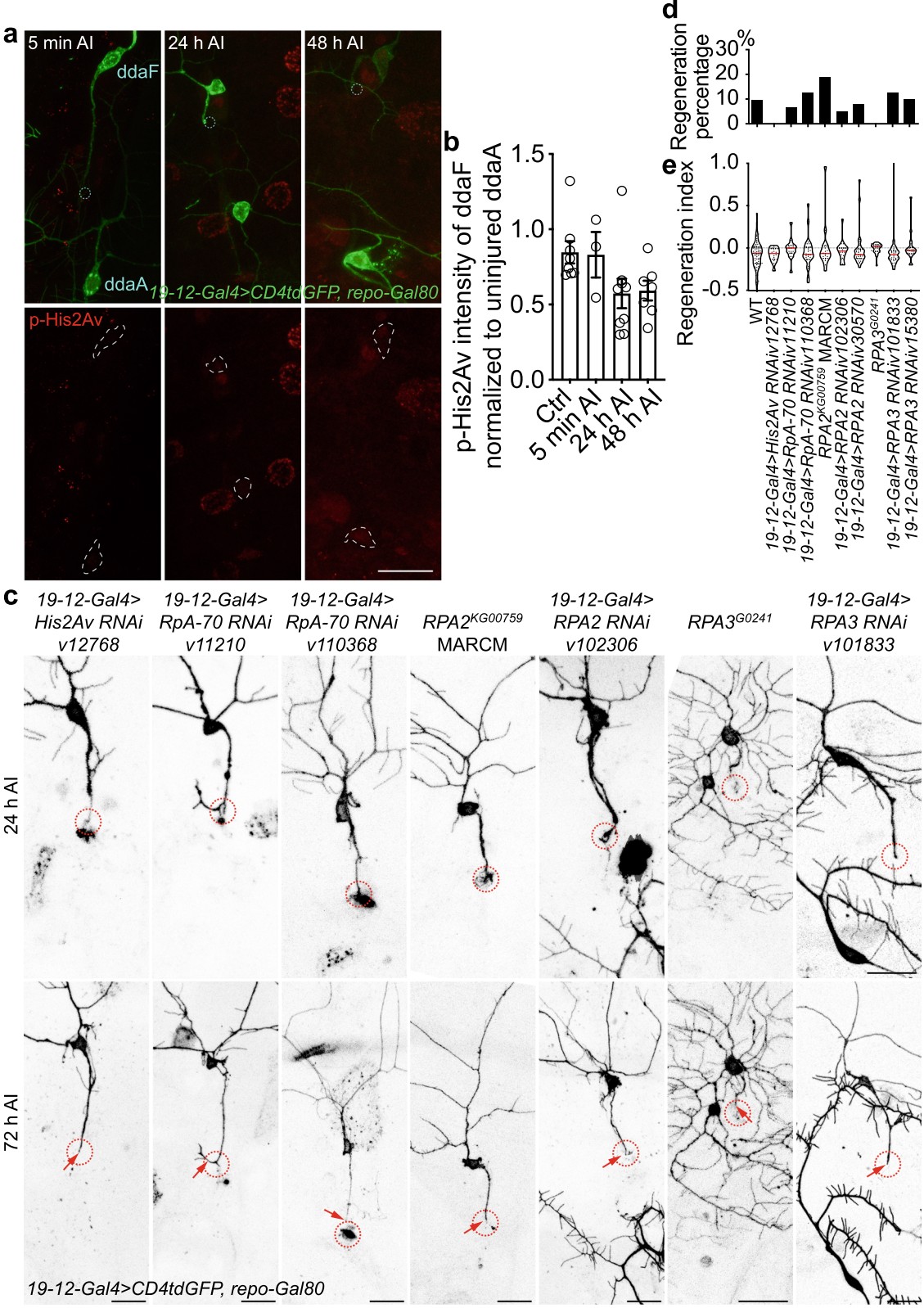

Second, we performed immunostaining for Atr, Atrip, Chek1, and TopBP1 using mouse dorsal root ganglion (DRG) tissues, and found that all of these proteins were expressed with/without sciatic nerve lesion (SNL) and that the expression of Atr and TopBP1 was modestly increased after injury (Supplementary Fig. 5b). Third, as extensive gene expression analyses had been performed in mouse and human DRGs, we thus queried the transcript level of Atr-Chek1 pathway members in a number of these databases[35,36], and found that they were all expressed in the mouse or human DRG, although at a low to medium level (Supplementary Fig. 5c). This is consistent with a potential homeostatic function of this pathway in non-dividing cells.

**Fig. 2 The Atr-Chek1 pathway regulates axon regeneration likely independent of DNA damage. a** The DNA damage marker, phosphorylated histone 2Agamma (p-His2Av), is not upregulated in injured class III da neurons (ddaF) at various time points, compared to the uninjured control (ddaA). The dashed teal circle marks the injury site and the cell bodies are outlined with dashed white lines. **b** The p-His2Av staining is quantified by normalizing the mean intensity in the injured class III da neuron (ddaF) to that of the neighboring uninjured class III da neuron (ddaA). No significant difference is observed at 5 min, 24 h, or 48 h AI. $N = 8$, 3, 9, and 7 neurons from 3 to 4 larvae. Data are presented as mean values +/− SEM. **c** Class III da neuron-specific expression of RNAis for His2Av, RpA-70, RPA2, or RPA3, $RPA2^{KG00759}$ mutant clones (with MARCM) or $RPA3^{G0241}$ mutants do not significantly increase axon regeneration. **d, e** Quantifications of class III da neuron axon regeneration with regeneration percentage (**d**) and regeneration index (**e**). $N = 72$, 14, 15, 24, 16, 20, 25, 24, 24, and 20 neurons from 4 to 20 larvae. The injury site is demarcated by the dashed circle. Arrow marks axon stalling. No statistical difference is detected by two-sided Fisher's exact test (**d**), one-way ANOVA followed by Dunnett's (**b**) and Holm–Sidak's test (**e**). Scale bar = 20 μm. Source data are provided as a Source data file.

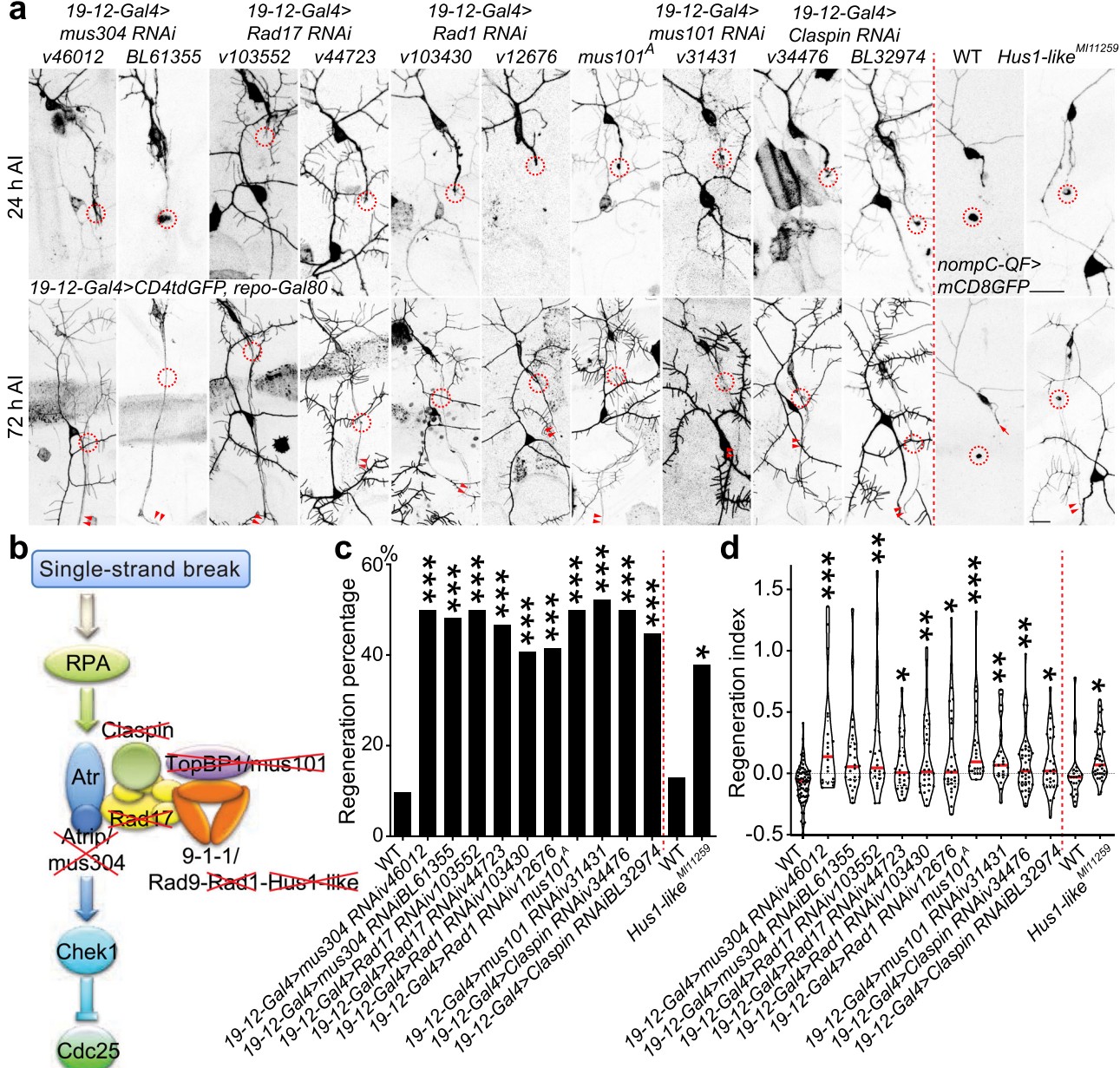

**Fig. 3 The Atr-associated checkpoint complex inhibits axon regeneration. a** TopBP1/mus101 and Hus1-like mutants, *mus101^A* and *Hus1-like^MI1259*, and class III da neuron-specific expression of Atrip/mus304 RNAis, Rad17 RNAis, Rad1 RNAis, TopBP1/mus101 RNAi, or Claspin RNAis increase axon regeneration. The injury site is demarcated by the dashed circle. Arrow marks axon stalling while arrowheads show the regrowing axon tips. **b** The single-stranded DNA-damage pathway mediated by Atr, Chek1, Cdc25, and the associated checkpoint complex. The factors marked by the red cross are tested for their potential role in axon regeneration. **c, d** Quantifications of class III da neuron axon regeneration with regeneration percentage (**c**) and regeneration index (**d**). $N = 72$, 20, 27, 24, 30, 32, 29, 24, 23, 44, 29, 23, and 37 neurons from 6 to 20 larvae. $P = 0.0002$, <0.0001, <0.0001, <0.0001, 0.0007, 0.0005, <0.0001, <0.0001, <0.0001, 0.0002, 0.0447. *$P < 0.05$, **$P < 0.01$, ***$P < 0.001$ by two-sided Fisher's exact test (**c**), one-way ANOVA followed by Dunn's test (**d**), two-tailed unpaired Student's *t*-test (**d**, $P = 0.0191$). Scale bar = 20 μm. Source data are provided as a Source data file.

**Blocking the Atr-Chek1 pathway promotes behavioral recovery and synapse regeneration.** Initiation of axonal regrowth is only the first step towards repairing lost connections. True repair requires that regenerating axons find their targets and re-form functional synapses. In general, this process of functional regeneration is poorly studied in the field. To our knowledge, this has never been documented in any *Drosophila* injury models[37]. To assess functional recovery in flies, we utilized a behavioral paradigm based on the larval gentle-touch response[38]—gently touching larval anterior segments (thoracic (T) segments and the first abdominal segment) with an eyelash elicits a set of stereotypical responses that are readily quantifiable (Supplementary Fig. 6b). We further subcategorized the intensity of the gentle-touch stimulus based on the contact area between the eyelash and the body segment: touch+, touch++, and touch+++, with the eyelash diameter covering <1/8, 1/8–1/4, or 1/4–1/2 of the segment, respectively (Supplementary Fig. 6a). Larvae showed a graded response according to the stimulation intensity (Fig. 4c, e, g and Supplementary Fig. 6d–f). As shown previously, the mechanosensitive class III da neurons mediate gentle-touch sensation[39]. We further found that there is a segment-wise somatosensory map for gentle-touch: class III da neuron axons project into the VNC (ventral nerve cord) in an anterior-posterior pattern, that is, axons from the T1 segment constitute the anterior-most T1 bundle within the VNC (Fig. 4a), and that injuring the class III da neuron axon bundle at T1 or T2 in the VNC (Fig. 4a) led to an impaired touch response specifically at segment T1 or T2, without affecting neighboring segments (Supplementary Fig. 6c).

Given this precision, we injured class III da neuron axons at the T1 & T2 bundles in the VNC, then specifically stimulated the T1, T2, T3, or A1 (abdominal) segment, and scored the touch response at 8, 24, and 48 h AI. We found that in WT control, axon injury in the VNC resulted in failed gentle-touch response at the T1 & T2 segments without affecting the T3 & A1 segments, when tested at 8 h AI (Fig. 4c, e, g and Supplementary Fig. 6d–f, Supplementary Movie 1). This defect persisted at 48 h AI (Fig. 4c, e, g and Supplementary Movie 2), with 0% and 7.3% of the larvae displayed behavioral recovery at 24 and 48 h AI (Fig. 4b, d, f and "Methods"), respectively. We went on to assess the behavioral outcome after inhibiting the Atr-Chek1 pathway focusing on mei41, Rad17, and mus101, all of which showed a strong axon regeneration phenotype. Strikingly, after knocking down *Rad17* in class III da neurons, in *mei41$^{29D}$* mutants and in particular in the *mus101* mutant—*mus101$^A$*, we observed substantial behavioral recovery as early as 24 h AI (26%), which continued to improve at 48 h AI (43.5%) (Fig. 4b, d, f and Supplementary Movie 3–6). The response score was also statistically improved with the touch+++ stimulation (Fig. 4c, e, g). In the meantime, the behavioral response in the uninjured T3 and A1 segments were comparable among the different genotypes (Supplementary Fig. 6d–f). These results demonstrate that inhibiting the Atr-Chek1 pathway is not only beneficial for axon regrowth, but also promotes functional regeneration.

In order to assess synapse regeneration, we first sought to confirm that class III da neurons form cholinergic synapses in the CNS. We found that class III da neurons were co-labeled by *nompC-QF > mCD8GFP* (the class III da neuron marker) and *ChAT-Gal4 > CD4TdTomato* (Supplementary Fig. 6g), which marks essentially all cholinergic neurons[40]. We used synaptotagmin-GFP (syt.eGFP/syt)[41] to mark class III da neuron presynapses (Fig. 5b), which were tightly opposed by postsynaptic cholinergic receptors labeled by α-bungarotoxin (α-BTX)[42] (Fig. 5a). We then ablated the class III da neuron axon bundles on one side of the VNC, which retracted out of the neuropil within 8 h AI (Fig. 5c). At 24 h AI, WT axons rarely regrew into

the neuropil, displaying retraction bulb-like structures (Fig. 5c). We subsequently focused on the *mus101$^A$* mutants, as they demonstrated the most robust behavioral recovery. In contrast to WT, *mus101$^A$* mutant class III da neurons not only exhibited extensive axon regeneration (~50%) back into the neuropil, but also increased the percent of regenerating axons containing syt puncta, indicative of synapse reformation (Fig. 5c, d, arrowheads).

**Atr functions downstream of Piezo and Nos in regulating axon regeneration.** Since Atr is not activated by DNA damage during neuroregeneration, what then, is the trigger? We speculated that mechanical stimulus may be the culprit, based on several lines of evidence. First, mechanical stress such as osmotic stress is capable of relocating and activating Atr, which then phosphorylates Chek1[16]. Second, during axon regeneration, mechanical force is reported to stimulate the MS ion channel Piezo, leading to local calcium elevation in the growth cone and a signaling cascade mediated by Nos to inhibit regeneration[17]. Third, removal of Piezo or Nos promotes class III da neuron axon regeneration to a similar extent as Atr/mei41 LOF. Fourth, we found that *PiezoKO* also increased class I da neuron axon regeneration as well as *Atr/mei41* mutants (Supplementary Fig. 2f–h, arrowheads). Last, Piezo can be activated by osmotic stress[18]. Therefore, we hypothesized that Piezo transduces the mechanical signal elicited during neuroregeneration, to trigger activation of the Atr-Chek1 pathway.

As a first step to testing this hypothesis, we sought to determine whether Atr's response to osmotic stress is Piezo-dependent. We first reproduced the hypotonic stress-induced Atr relocalization/clustering phenomenon previously shown in HeLa cells[16]. Using WT HEK293T cells, we found that exogenously expressed FLAG-tagged human ATR also formed clusters in the nucleoli under the hypotonic condition (Fig. 6a, dashed circles). However, in PIEZO1 knockout HEK293T cells (*PIEZO1KO*)[43], ATR clusters were much less abundant. Even if they were induced, they appeared smaller in size and showed reduced fluorescence intensity (Fig. 6a–c). We also repeated this experiment using an antibody that recognizes the endogenous ATR, together with a nucleoli marker—nucleolin. We found that a 15 min exposure to hypotonic stress-induced significantly more ATR clustering in WT cells compared to *PIEZO1KO*, as quantified by the "number of ATR puncta per cell" and the "percent of ATR puncta colocalized with nucleolin" (Supplementary Fig. 7a–c). This result demonstrates that the ATR clustering phenomenon is consistent between the endogenous and ectopically expressed ATR. We then went on to determine whether Atr clustering relies on Nos, as it is downstream of Piezo. We pharmacologically manipulated NOS, and found that inhibiting NOS with 1400 W dihydrochloride[44] attenuated the ATR clustering induced by hypotonic stress, whereas activating NOS with histamine[45] rescued the reduced ATR clustering in *PIEZO1KO* (Fig. 6d–f). These results indicate that the mechanical stress-induced Atr relocalization relies on the presence of Piezo and Nos, and that Piezo can function as a mechanoreceptor underlying Atr's response to mechanical stimulus.

Second, we performed genetic interaction and epistasis analyses to determine the relationship between Atr/mei41 and the Piezo-Nos pathway. Genetic interaction analyses showed that while transheterozygotes of *mei41$^{29D}$/+; PiezoKO/+* did not show a regeneration phenotype, *mei41$^{29D}$/+; Nos$^{Δ15}$/+* significantly promoted class III da neuron axon regeneration, similar to homozygous mutants of *mei41$^{29D}$, PiezoKO*, or *Nos$^{Δ15}$* (Fig. 7a–c). This result indicates that Atr/mei41 and Piezo-Nos function in the same genetic pathway, and further suggests that although Atr/mei41 does not associate with Piezo per se, it

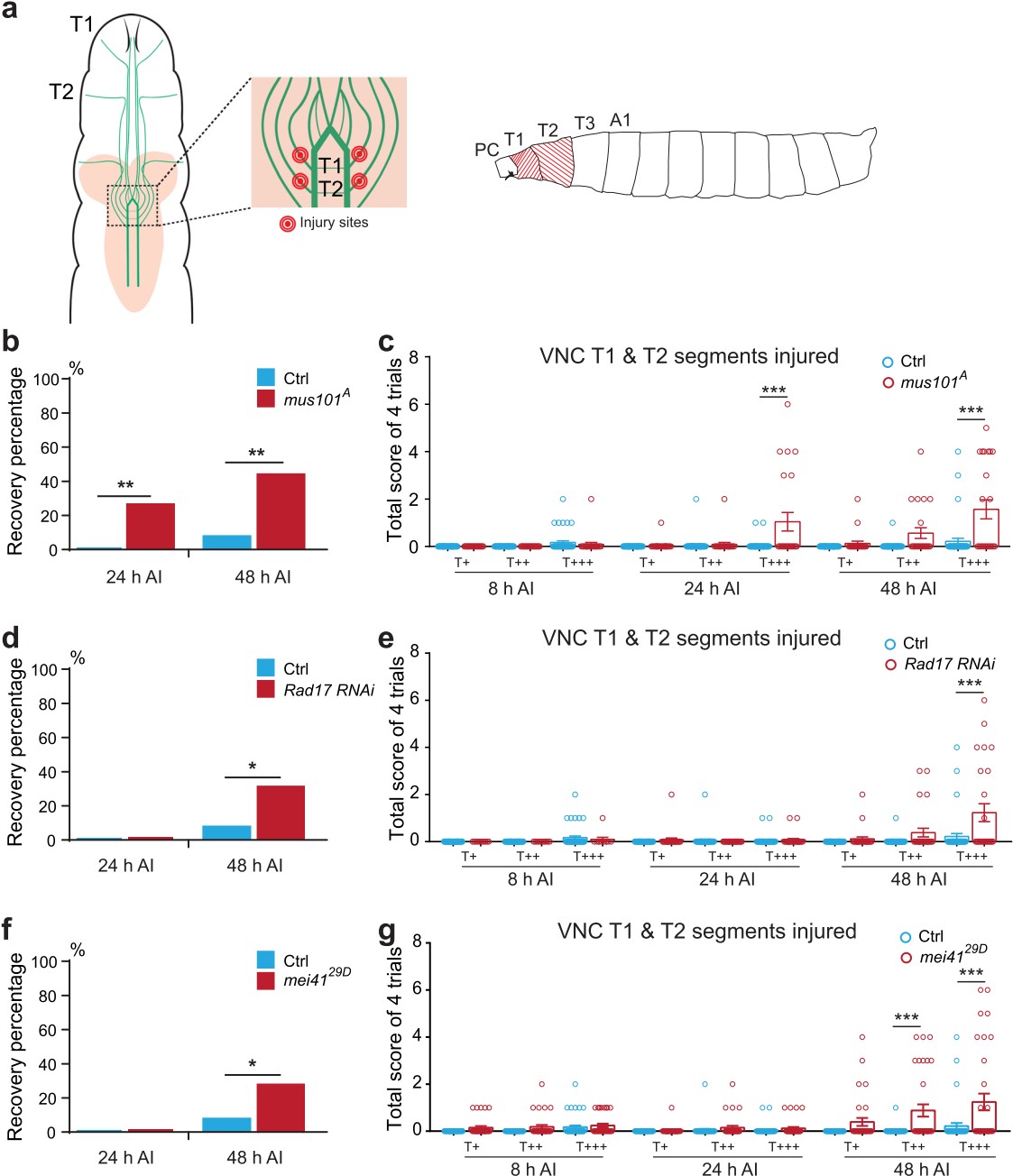

**Fig. 4 Inhibiting components of the Atr-associated checkpoint complex promotes behavioral recovery after CNS injury in flies. a** Class III da neuron axon projection map in the VNC and the VNC injury paradigm. There is a segment-wise somatosensory map for gentle-touch: class III da neuron axons project into the VNC in an anterior-posterior pattern. Axons from the T1 segment constitute the anterior-most T1 bundle within the VNC. The T1 and T2 axon bundles are injured by targeting the nerve bundles right before they enter the commissure region, as marked by the red dots. Gentle-touch response is then performed by stimulating the T1 or T2 segments using an eyelash. A total of four trials are scored for each larva. PC, pseudocephalon; T, thoracic; A, abdominal. **b** $mus101^A$ mutants show enhanced gentle-touch response after VNC injury, as shown by the Recovery percentage. A larva is defined as showing recovery if the scores from at least two of the four trials are 1 or above. While WTs largely fail to respond even at 48 h AI, significantly more $mus101^A$ mutants show recovery as early as 24 h AI. $P = 0.0013, 0.001$. **c** Gentle-touch response scores at 8, 24, and 48 h AI with various stimulation intensities. $Mus101^A$ mutants display significantly higher recovery, especially with the T+++ stimulus. **d** Class III da neuron-specific knockdown of Rad17 mildly increases Recovery percentage at 48 h AI. $P = 0.0176$. **e** Class III da neuron-specific knockdown of Rad17 improves response scores at 48 h AI. **f** $mei41^{29D}$ mutation mildly increases Recovery percentage at 48 h AI. $P = 0.0277$. **g** $mei41^{29D}$ mutation improves response scores at 48 h AI. $N = 41$ larvae for Ctrl, 23 for $mus101^A$, 33 for $mei41^{29D}$, 11, 26, and 26 for $Rad17$ RNAi at 8, 24, and 48 h. Data are presented as mean values $+/-$ SEM. $*P < 0.05, **P < 0.01, ***P < 0.001$ by two-sided Fisher's exact test (**b**, **d**, and **f**), one-way ANOVA followed by Tukey's test (**c**, **e**, and **g**). Source data are provided as a Source data file.

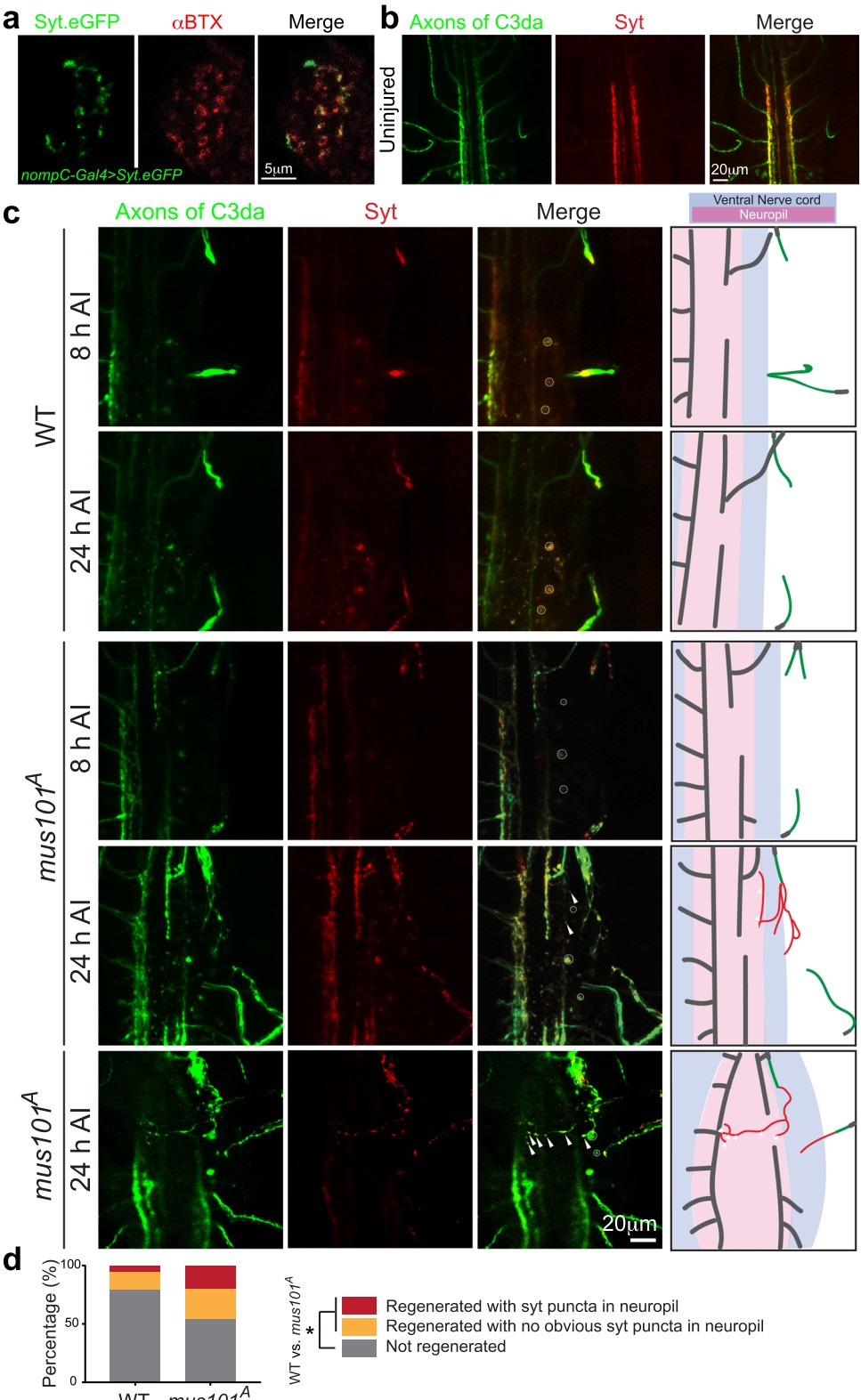

appears to closely interact with Nos. Moreover, our epistasis analysis showed that (1) class III da neuron-specific over-expression of Chek1/grp reduced the enhanced regeneration phenotype in *PiezoKO*; (2) overexpression of Chek1/grp also attenuated the regeneration enhancement in *Nos*$^{\Delta 15}$ mutants; (3) on the other hand, class III da neuron-specific overexpression of mPiezo1-TriM, an over-activating mutant of mouse Piezo1 that reduces axon regeneration when overexpressed in class IV da

neurons[17], did not significantly reduce the enhanced regeneration in *mei41*$^{29D}$ mutants; (4) overexpression of Nos, which inhibits axon regeneration in class IV da neurons[17], also failed to attenuate the regeneration phenotype in *mei41*$^{29D}$ mutants (Fig. 7a–c). These data collectively suggest that Atr/mei41 operates downstream of Piezo-Nos and that Atr/mei41 overrides the regeneration phenotype that results from the removal of Piezo or Nos.

**Fig. 5 Inhibiting the Atr pathway promotes synapse regeneration in flies. a** Class III da neurons form cholinergic synapses in the VNC. Synaptotagmin-GFP (syt.eGFP/syt) marks class III da neuron presynapses, which are tightly opposed by postsynaptic cholinergic receptors labeled by α-bungarotoxin (α-BTX). Scale bar = 5 μm. **b** In uninjured class III da (C3da) neurons (marked in green), syt puncta (marked in red) are enriched at the presynaptic terminals within the neuropil. Scale bar = 20 μm. **c** *Mus101^A* mutants show enhanced axon regeneration and synapse reformation in the CNS. Class III da neuron axon bundles on one side of the VNC are ablated (dashed circles), resulting in the retraction of axons out of the neuropil within 8 h AI. At 24 h AI, WT axons rarely regrow into the neuropil, displaying retraction bulb-like structures. *Mus101^A* mutant class III da neurons not only exhibit extensive axon regeneration back into the neuropil, but also increase the percent of regenerating axons containing syt puncta at the terminals (arrowheads), indicative of synapse reformation. Two examples of *mus101^A* mutants are shown. The schematic drawings depict the VNC (blue), neuropil (pink), uninjured axons (black), retracted axons (green), and regenerating axons (red). Scale bar = 20 μm. **d** Quantification of axon and synapse regeneration. *N* = 34 and 35 axon bundles from 8 and 10 larvae. The percent of regenerated axons increases from 21% in WT to 47% in *mus101^A* mutants, *P* = 0.0406. **P* < 0.05 by two-sided Fisher's exact test. The percent of regenerated axon containing syt puncta is also increased in *mus101^A* mutants. Source data are provided as a Source data file.

Third, our results that Atr/mei41 genetically interacts with Nos and that Nos activity is required for the mechanical stress-induced Atr clustering suggest that Piezo feeds into the Atr-Chek1 pathway through NO (nitric oxide) signaling, which would be consistent with a previous finding that NO promotes p53 nuclear retention in an Atr-dependent manner[46]. To directly visualize NO propagation, we performed NO imaging using the fluorescent NO dye DAF-FM diacetate[47,48]. While NO production was rarely detected in uninjured class III da neurons (Fig. 7f and Supplementary Fig. 8), we observed obvious fluorescence signal around the axon tip, along the axon, and in the cell body in 62.5% of the WT class III da neurons at 24 and 48 h AI (Fig. 7d, f). As a negative control, we found that the signal of the NO dye was drastically reduced in the Nos mutant—*Nos^Δ15* (Fig. 7d, f, g). Importantly, NO production was similarly abolished in *PiezoKO* (Fig. 7e–g). In addition, we found that NO signal was also present in class IV da neurons after injury, but to a lesser extent with the percent of NO⁺ neurons lower than that of class III da neurons (Supplementary Fig. 8b, c). These results, together with the genetic analyses, suggest that NO functions as a key messenger that links the activation of the Piezo channel during axon regeneration to the downstream Atr-Chek1 pathway.

**Pharmacological and genetic inhibition of the Atr-Chek1 pathway promotes axon regeneration in mammalian neurons in vitro and in vivo.** Having established that hATR and hCHEK1 inhibit axon regeneration in flies, we hypothesized that Atr-Chek1 may also function as regeneration inhibitors in mammals. First, we tested this hypothesis using an injury paradigm in cultured neurons in vitro based on a microfluidic device[49,50]. In brief, embryonic day 18 (E18) rat DRG neurons were cultured in a microfluidic chamber that separates the neurites from the soma. The neurites in the terminal chamber were removed by vacuum aspiration at 7 days in vitro (DIV7) and regeneration was assessed at various time points after injury. We tested the efficacy of Atr and Chek1 inhibitors in promoting axon regeneration. The Atr-Chek1 network is a key mediator of DDR, and inhibiting DDR has become an attractive concept in cancer therapy. To date, pharmacological inhibitors for Atr and Chek1 have already entered anti-cancer clinical trials either as stand-alone agents or combined with radio- or chemotherapy[51,52]. We thus tested two pharmacological inhibitors of Atr—AZD6738 and VE-822 (NCT02157792, NCT02223923, and NCT02264678) and the Chek1 inhibitor MK-8776[53,54] (NCT01870596). We found that inhibiting Atr with AZD6738 or VE-822, or inhibiting Chek1 with MK-8776 all modestly promoted axon regeneration at 18 h AI (Fig. 8a). The regrown neurite coverage areas ("Methods") were modestly but significantly increased in drug-treated neurons, as compared to the vehicle-treated controls (Fig. 8b). The AZD6738-dependent neurite regrowth enhancement was already apparent at 5 h AI (Fig. 8c). We next asked whether Chek1

inhibition might promote regeneration in vivo. To address this question, we injected the Chek1 inhibitor MK-8776 directly into fly larvae immediately after axonal injury and found that this compound significantly increased class III da neuron axon regeneration (Fig. 8d–f, arrowheads), compared to the vehicle control (Fig. 8d–f, arrow).

Third, to analyze axon regeneration in mammals in vivo, we utilized the sciatic nerve lesion (SNL) model in adult mice. To generate sensory neuron-specific Atr conditional knockout (*Atr cKO*), we bred mice with *Advillin-CreER; Atr^{fl/fl}* alleles and induced Cre-mediated recombination with tamoxifen (TAM) injection ("Methods"). Regenerating sensory axons were identified by SCG10 immunostaining[55] (Fig. 8g). We found that the extension of SCG10⁺ axons was significantly increased (~60%) in *Atr cKO* compared to control animals at SNL Day 3 (Fig. 8g, h and Supplementary Fig. 11a). On the other hand, overexpression of human CHEK1 in DRG neurons (via AAV2) reduced axon regeneration after SNL (Fig. 8i, j and Supplementary Fig. 11a). To summarize, these results suggest that the Atr-Chek1 pathway also functions intrinsically in neurons to inhibit axon regeneration in mammals, a process that may be evolutionarily conserved, and that the anti-cancer drugs targeting Atr-Chek1 may be repurposed for treating neural injury.

**Piezo exerts greater inhibition of axon regeneration on soft versus stiff substrates.** To begin to understand how Piezo becomes activated during axon regeneration, we asked if external factors of the microenvironment, specifically the stiffness of the extracellular matrix, might affect axon outgrowth in the presence or absence of Piezo1. We approached this question by culturing adult mouse DRG neurons on polyacrylamide (PAA) hydrogel substrates of varying stiffnesses and measured the total neurite lengths. It was previously reported that DRG neurites have maximal extensions and traction forces at an "intermediate" stiffness (elastic modulus of 1000 Pa)[56]. Therefore, we created hydrogels of shear moduli: 0.1, 0.3, 1.0, 5.0, and 30.0 kPa (with 0.1 and 30.0 kPa being at the extreme ends of "soft" and "stiff") to catch the range of substrate stiffnesses in which Piezo is most likely to perform its inhibitory role. We hypothesize that an "extremely" stiff environment (30.0 kPa) will negate any Piezo-mediated difference in traction force whereas the lower and intermediate gels will result in a difference, thus indicating the stiffness at which Piezo is active. To generate sensory neuron-specific Piezo1 conditional knockout (*Piezo1 cKO*), we bred mice with *Advillin-CreER; Piezo1^{fl/fl}* alleles and induced Cre-mediated recombination with TAM injection ("Methods"). Adult DRG neurons were dissociated and cultured onto our PAA hydrogels[57,58] for 38–40 h. Indeed, we found that there was no difference in total neurite growth between control and *Piezo1 cKO* groups when grown on 5.0 or 30.0 kPa hydrogels (Fig. 8k, l and Supplementary Fig. 11b). However, on 0.3 and 1.0 kPa

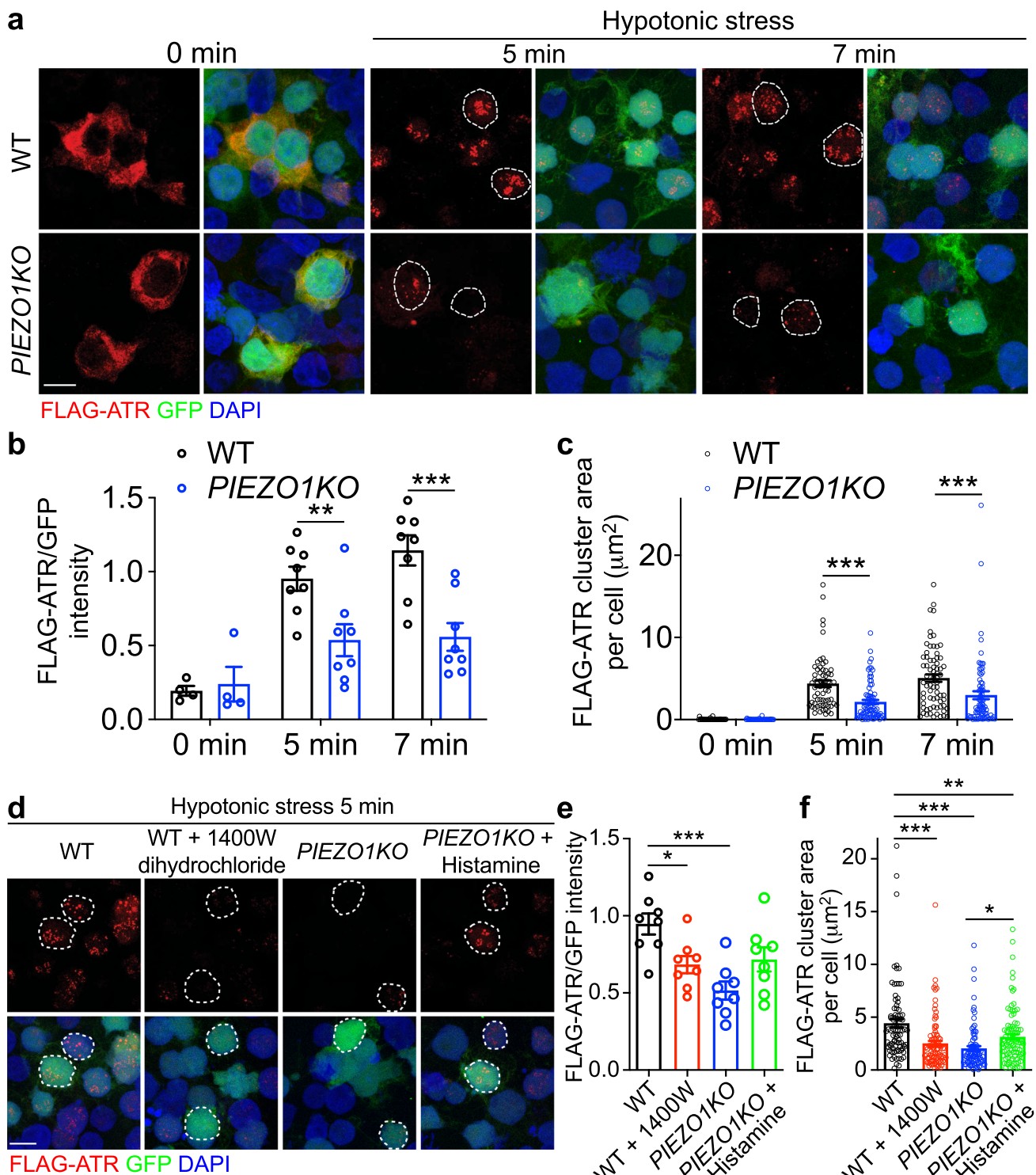

**Fig. 6 ATR's response to osmotic stress depends on PIEZO1 and NOS. a–c** Hypotonic stress-induced ATR clusters in the nucleus are attenuated in PIEZO1 knockout. **a** Exogenously expressed FLAG-ATR is present in the cytoplasm in both WT and *PIEZO1KO* HEK293T cells before treatment. 5 min or 7 min hypotonic stress induces robust clustering of FLAG-ATR in the nucleus in WT cells, which is much attenuated in the *PIEZO1KO* cells. Fewer cells produce the clusters. The clusters are smaller in size, fewer in number, and lower in intensity. The dashed circles outline the nucleus. Scale bar = 10 μm. **b** Quantification of the fluorescence intensity of FLAG-ATR normalized to GFP shows a reduction in *PIEZO1KO* cells. $N = 4$, 8, and 8 fields of view. **c** Quantification of the total area of FLAG-ATR clusters in the nucleus per cell is also reduced in *PIEZO1KO* cells. $N = 36$, 65, and 70 cells. **d–f** ATR clustering depends on NOS. **d** Hypotonic stress-induced ATR clusters in WT HEK293T cells are reduced by the NOS inhibitor 1400 W dihydrochloride, while histamine, a NOS activator, increases ATR clusters in *PIEZO1KO* cells. The dashed circles outline the nucleus. Scale bar = 10 μm. **e** Quantification of the fluorescence intensity of FLAG-ATR clusters. $N = 8$ fields of view. **f** Quantification of the total area of FLAG-ATR clusters in the nucleus per cell. $N = 86$, 94, 87, and 97 cells. Data are presented as mean values +/− SEM. \*$P < 0.05$, \*\*$P < 0.01$, \*\*\*$P < 0.001$ by two-way ANOVA followed by Sidak's test (**b** and **c**), one-way ANOVA followed by Tukey's multiple comparisons test (**e** and **f**). Source data are provided as a Source data file.

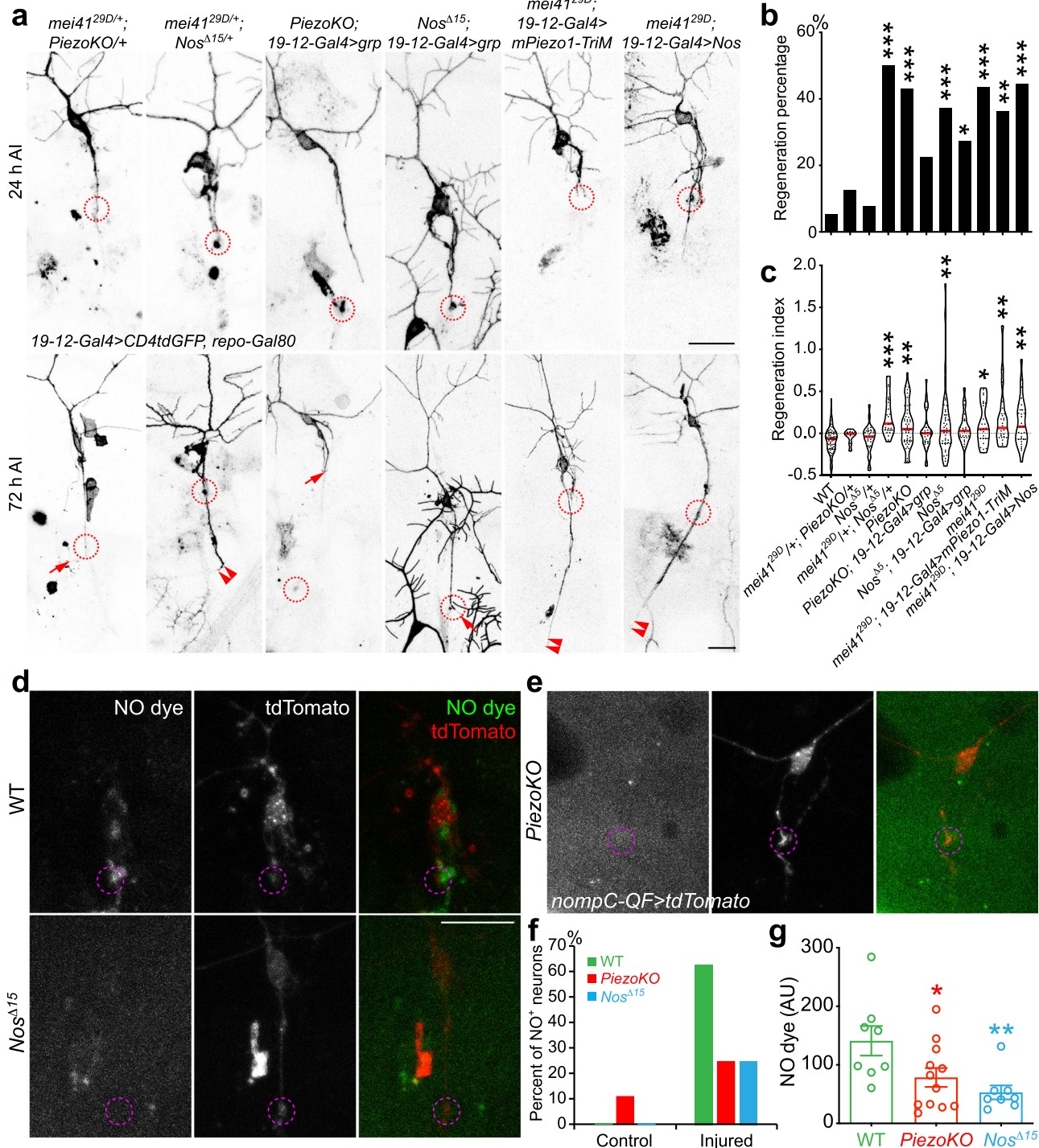

hydrogels, *Piezo1 cKO* DRG neurons exhibited significantly more total neurite length than control neurons (Fig. 8k, i and Supplementary Fig. 11b), suggesting that Piezo1 is likely to be activated on softer substrates. We also noticed that *Piezo1 cKO* neurons do not extend axons well on 0.16 kPa substrates, consistent with the reported role of Piezo1 in mediating axon outgrowth in *Xenopus*[59]. Worth mentioning, the elastic modulus, a measure of the tissue's resistance to deformation, ranges from 50 to 500 Pa for uninjured cortical tissues in rats, and it is even lower in injured tissues[60]. Therefore, it is possible that during axon regeneration, the local force between the axon tip and the environment falls into the range for Piezo activation. Given the enrichment of Piezo at the axon tip after injury[17], our findings

suggest that Piezo is thus capable of transducing the physical stimuli from the growth cones into the intracellular signaling cascades to slow down axon regeneration. Last, we tested *Atr cKO* DRG neurons and found that they showed a similar response on the stiffness assay, as the *Piezo1 cKO* neurons (Fig. 8m and Supplementary Fig. 11b). Specifically, while Piezo1 is activated at 0.3–1 kPa, Atr is more active at 0.3 kPa, suggesting that Atr is likely only one of the signaling branches downstream of Piezo1.

## Discussion

Mature neurons retain limited capacity to repair their injured nerve fibers after trauma, leading to poor functional recovery. To

**Fig. 7 Atr/mei41 functions downstream of Piezo and Nos in inhibiting axon regeneration, and NO imaging. a** Genetic interaction and epistasis analyses among *Piezo*, *Nos*, and *Atr/mei41*. While class III da neuron axons in *Nos$^{\Delta15}$* heterozygotes, or transheterozygotes of *PiezoKO* and *mei41$^{29D}$* (*mei41$^{29D}$/+*; *PiezoKO/+*) behave similarly to WT, significant enhancement of regeneration is observed in transheterozygotes of *Nos$^{\Delta15}$* and *mei41$^{29D}$* (*mei41$^{29D}$/+*; *Nos$^{\Delta15}$/+*). Class III da neuron-specific overexpression of grp in *PiezoKO* or *Nos$^{\Delta15}$* mutants reduces their regeneration enhancement phenotype. On the other hand, Class III da neuron-specific overexpression of mPiezo-TriM or Nos fails to suppress the regeneration enhancement in *mei41$^{29D}$* mutants. The injury site is demarcated by the dashed circle. Arrow marks axon stalling while arrowheads show the regrowing axon tips. **b, c** Quantifications of class III da neuron axon regeneration with regeneration percentage (**b**) and regeneration index (**c**). N = 37, 8, 26, 24, 49, 31, 43, 22, 23, 22, and 27 neurons from 3 to 14 larvae. P = 0.4524, >0.9999, <0.0001, <0.0001, 0.0687, 0.0009, 0.0428, 0.0006, 0.0037, 0.0004. **d–g** NO imaging in WT, *Nos$^{\Delta15}$* mutants, or *PiezoKO* at 48 h AI. **d** NO production is detected by DAF-FM diacetate. While in WT, NO is present around the injured axon tip, along the axon, and in the cell body, the fluorescence signal is drastically reduced in *Nos$^{\Delta15}$* mutants which lack the NO producing enzyme. **e** The NO fluorescence signal is similarly reduced in *PiezoKO*. The injury site is demarcated by the dashed circle. **f** NO fluorescence signal is rarely detected in uninjured control class III da neurons. At 48 h AI, 62.5% of the WT class III da neurons show obvious NO fluorescence signal, compared to 25% in *PiezoKO* or *Nos$^{\Delta15}$* mutants. **g** The mean NO fluorescence intensity measured at the growth cone tip is also significantly reduced in *PiezoKO* or *Nos$^{\Delta15}$* mutants. N = 8, 12, and 8 neurons from 3 to 4 larvae. Data are presented as mean values +/− SEM. *P < 0.05, **P < 0.01, ***P < 0.001 by two-sided Fisher's exact test (**b**), one-way ANOVA followed by Holm–Sidak's test (**c**), or Dunnett's test (**g**). Scale bar = 20 µm. Source data are provided as a Source data file.

overcome this regeneration failure, numerous efforts have been made to increase intrinsic axon regrowth and/or remove extrinsic obstacles[1–4]. However, we still have a limited understanding of how injured neurons integrate extrinsic information with the intrinsic signaling pathways to make the decision to regenerate, stall, retract or die. In this study, using a *Drosophila* sensory neuron injury model, we identified the Atr-Chek1 pathway as a neuron-intrinsic negative regulator of axon regeneration. We hypothesize that during axon regeneration, the growth cone physically interacts with the environment such as the glial cells, resulting in the activation of the mechanosensitive ion channel Piezo at the growth cone tip; opening of the Piezo channels leads to local calcium influx and the activation of Nos, which then produces NO; NO functions as a second messenger and propagates to the nucleus where it activates Atr and its associated complex; Atr then phosphorylates and activates Chek1, which phosphorylates and inactivates Cdc25, inhibiting its ability to dephosphorylate and activate Cdk1; the phosphorylated and inactive Cdk1 impinges on downstream effectors, causing regeneration failure (Supplementary Fig. 9).

The Piezo-Nos-Atr cascade thus behaves as a regeneration brake in neurons, which is capable of sensing the extrinsic cues in the local microenvironment, processes and transduces these signals to a kinase circuit originally known to respond to DNA damage, and then potentially rewires the circuit to instruct cellular events such as cytoskeleton reorganizations to curtail regeneration. Further downstream, we speculate that Cdk1 may phosphorylate multiple substrates and thus engage multiple pathways. For example, Cdk1 activates FOXO[61], and Daf16/FOXO is inhibited by the insulin/IGF1 receptor DAF-2 during the age-dependent decline of axon regeneration in *C. elegans*[62]. Cdk1 also phosphorylates Nde1[63], which regulates dynein-dependent transport, another process that is important for axon regeneration[64], associating with the regulation of the cytoskeleton. Cytoskeleton dynamics, in particular microtubule and actin, have been well documented as key players in axon regeneration[65]. The Piezo-Nos-Atr machinery is not restricted to the regeneration-incompetent class III da sensory neurons in flies, as was confirmed by the presence of Atr/mei41 in other types of da neurons, and by the enhanced axon regeneration of class I da neurons in *PiezoKO* or *Atr/mei41* mutants, mammalian DRG neurons after pharmacological inhibition of Atr or Chek1, or in *Atr cKO*. We propose that this may be a mechanism adopted by injured neurons in general, as a wait-and-see strategy, allowing them the opportunity to sample the environment, assess the intrinsic status, and decide whether to regenerate. A lot of intrinsic factors have been identified to regulate the ability of axon regeneration, and some of them are pro-regenerative while

others are inhibitory to regeneration[5]. In our view, whether one neuron can regenerate its axon is determined by the synergistic effect of various factors. Furthermore, there may be a tug-of-war in all neuronal types after injury, between factors that inhibit regeneration such as Piezo-Nos-Atr, and those that promote regeneration. In class III da neurons, because of their limited intrinsic regenerative potential, the activation of Piezo-Nos-Atr is sufficient to suppress further regeneration. However, in class IV da neurons, the endogenous regenerative drive is high enough to override this blockade. It is worth noting that DmPiezo is expressed in both class III and IV da neurons[17], and that it is likely not the primary cause of the differences in regeneration. The pro-regenerative machinery may be determined by differentially expressed calcium channels among various neuronal subtypes[66], whereas the Piezo-Atr mediated inhibition may be a more general mechanism in neurons. In addition, while the Piezo-NO-Atr pathway is active in different types of neurons, its strength, which contributes to the regeneration–inhibition ability of this pathway, may vary. This is consistent with the different regeneration abilities between class III and class IV da neurons. The regeneration enhancement phenotype observed in *Atr/mei41* mutants is comparable to that of *PiezoKO* in flies, and both Piezo and Atr are shown to not only inhibit axon regeneration in fly larvae but also in adult sensory neurons in mice[17]. These findings raise the possibility that the Atr pathway plays an evolutionarily conserved role in regulating regeneration, which warrants further investigation.

The involvement of DNA damage in neural injury and regeneration is an emerging concept, but not well explored. p53 was reported to be required for facial nerve regeneration in mice[67]. Whereas inhibition of Poly (ADP-ribose) polymerases (PARPs), which are involved in DNA repair, promotes axon regeneration in worms[68], no positive effect was observed in mice[69]. An outstanding question is whether axon injury induces DDR, and if so, how. More specifically, what type of DNA damage is potentially involved? Our results, on the other hand, favor the hypothesis that DNA damage may not play a prominent role at least in fly sensory neurons after injury. However, it is worth noting that we are not able to fully rule out the involvement of DNA damage, given our lack of ability to specifically detect single-stranded DNA breaks in injured neurons. Moreover, NO itself has been reported to act as a mutagen[70]. Thus, future studies are warranted to determine the link between NO and the Atr complex, and to assess additional NO targets. Another intriguing question is whether other cell types in the microenvironment, for example, the glial cells, undergo injury-induced DDR, and if so, how that may affect the regeneration outcome.

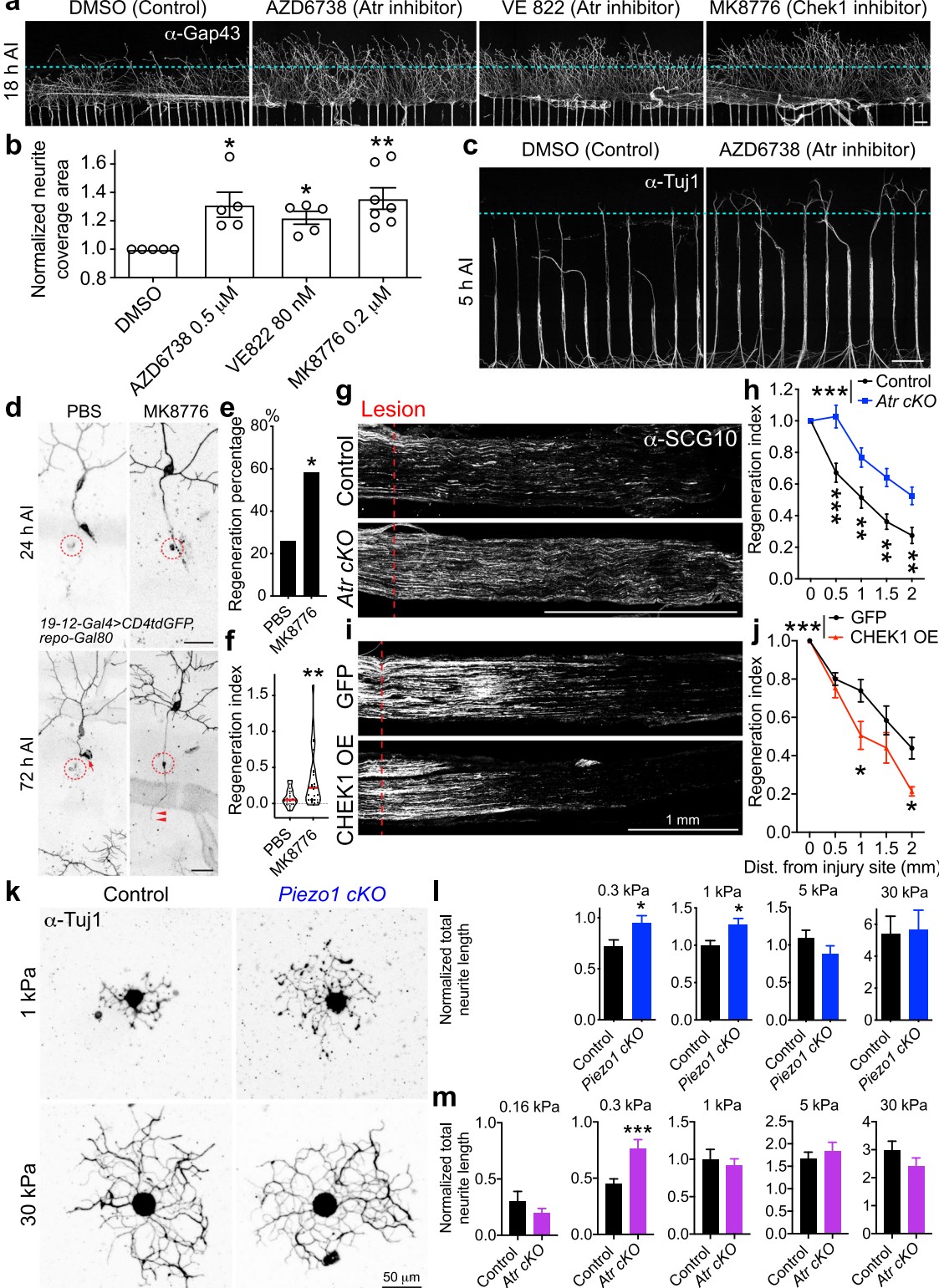

Activation of the Atr kinase is well known as the initial response to DNA damage. But interestingly, the Atr-Chek1 pathway is also involved in DNA-damage-independent functions. Mechanical stress activates Atr at the nuclear envelope to modulate chromatin dynamics and nuclear envelope plasticity[16]. Functions of the cytoplasmic Atr are associated with the centrosome, mitochondria, and cytoskeleton[71]. In addition, patients with Seckel syndrome due to ATR mutations, suffer severe nervous system malfunctions including microcephaly, defective neurodevelopment, and cognitive impairment[72], emphasizing a crucial neural function of Atr. Our work identified the missing mechanoreceptor underlying Atr's response to mechanical stimuli, which is the mechanosensitive ion channel Piezo. Together with previous work demonstrating Piezo's inhibitory role during axon regeneration, we have uncovered a route through which the mechanical force at the growth cone is sensed, encoded, and

**Fig. 8 Inhibition of the Atr pathway by pharmacological inhibitors or conditional knockout promotes mammalian DRG neuron axon regeneration in vitro and in vivo, and axon outgrowth on substrates of differing stiffness. a–c** Pharmacological inhibition of Atr or Chek1 modestly enhances axon regeneration of rat embryonic DRG neurons cultured in a microfluidic chamber, when applied after injury. **a** Inhibiting Atr with AZD6738 (0.5 μM), Chek1 with VE-822 (80 nM), or MK-8776 (0.2 μM) accelerates axon regeneration when imaged at 18 h AI. The axons are labeled with α-Gap43 staining. The dashed line marks the front of the axon tips in Control. **b** The axon coverage area is measured and normalized to the total width of the microgrooves. The values from the inhibitor-treated groups are further normalized to the corresponding DMSO vehicle control group in the same experiment. $N = 5, 5, 5$, and 7 experiments. **c** Enhanced axon regeneration is visible at 5 h AI when Atr is inhibited with AZD6738. The axons are labeled with α-Tuj1 staining. Scale bar = 100 μm. **d** Injection of the Chek1 inhibitor MK-8776 (final concentration: ~0.3 μM) into fly larvae right after injury enhances class III da neuron axon regeneration, compared to the PBS injected control. Arrow marks retracted axon tip and arrowheads mark the regenerating axon. Scale bar = 20 μm. **e, f** Quantifications of class III da neuron axon regeneration with regeneration percentage (**e**) and regeneration index (**f**). $N = 23$ and 24 neurons from 4 larvae. $P = 0.0392, 0.0052$. (**g, h**) *Atr cKO* enhances sensory axon regeneration in vivo. Analysis of regeneration of sensory axons by SCG10 immunostaining at SNL D3. Shown are sample images of regenerating sensory axons identified by SCG10 (**g**) and quantification (**h**). SCG10 immunofluorescence intensity was measured at different distal distances and normalized to that at the lesion site as the regenerative index. Dashed line marks the lesion site. Scale bar = 1 mm. $N = 12$ mice for each genotype. **i, j** Human CHEK1 overexpression in DRG neurons reduces sensory axon regeneration in vivo. Scale bar = 1 mm. $N = 5$ mice for each genotype. **k, l** *Piezo1 cKO* increases adult DRG neuron axon outgrowth on hydrogels of 0.3 and 1 kPa, but not 5 or 30 kPa. **k** Representative images of DRG neurons (stained with the α-Tuj1 antibody) grown on substrates of different stiffness. Scale bar = 50 μm. **l** Quantification of total neurite length normalized to that of the Control. All data points are normalized to the mean of the Control at 1 kPa. $N = 37, 41, 67, 54, 31, 23, 29$, and 34 neurons. $P = 0.0188, 0.0104, 0.8871, 0.1567$. **m** *Atr cKO* increases adult DRG neuron axon outgrowth on hydrogels of 0.3 kPa, but not 0.16, 1, 5, or 30 kPa. $P = 0.2684, 0.0005, 0.6222, 0.4928, 0.2176$. $N = 28, 34, 73, 96, 161, 90, 78, 68, 26$, and 22 neurons. Data are presented as mean values +/− SEM. *$P < 0.05$, **$P < 0.01$, ***$P < 0.001$ by two-sided Fisher's exact test (**e**), one-way ANOVA followed by Holm–Sidak's test (**b**), two-tailed unpaired Student's *t*-test (**f, i**, and **m**), or Two-way ANOVA followed by Sidak's test (**h** and **j**). Source data are provided as a Source data file.

transduced to the nucleus to elicit a signaling cascade governing cellular events such as regeneration. It is worth mentioning that our work does not rule out the possibility that Atr itself may also be mechanosensitive. ATR has been shown to contain HEAT (huntingtin, elongation factor 3, A subunit of protein phosphatase 2A and TOR1) repeats[73], which can behave as an elastic connector, potentially capable of sensing mechanical forces[74].

Accumulating evidence has shown that axons and growth cones generate mechanical forces when they are interacting with the substrates during growth[75,76]. During the interaction with neurons, stiffness of the environment is an important aspect that determines the extent of mechanical forces which eventually impact neurons. Piezo, as an ion channel directly gated by mechanical forces, senses a certain range of substrate stiffnesses in both neural and non-neural cells. In other words, substrates with stiffness within a certain range can activate Piezo channels[59,77,78]. In our previous work, we have demonstrated that Piezo is activated in terminals of axotomized axons in *Drosophila*[17], suggesting that the stiffness of *Drosophila* neural environment is capable of activating Piezo, and that mechanical forces exist between the axotomized axons and the environment. In the current work, we found that, on a substrate with a close-to-in vivo stiffness, neurite outgrowth is indeed mediated by Piezo. However, Piezo did not function when the substrate stiffness is much greater than the in vivo range. These results further support our hypothesis that a sustained signal for Piezo channels would come from the interaction between axons and the in vivo neural environment, either in *Drosophila* or mice. As no existing approaches have been demonstrated to have enough resolution of directly and precisely detecting the mechanical forces in vivo in the regenerating axon tips, we believe that the evidence that we have provided, together with our previous publication, has demonstrated an important link between Piezo activation and substrate stiffness: (1) Piezo is enriched in the axon tips after injury in vivo[17]; (2) Piezo is activated and regulates the calcium signaling in axons during regeneration in vivo[17]; (3) in vitro, Piezo can impact neurite outgrowth when the stiffness of substrate is close to that of the neural environment in vivo, but not when it is physiologically irrelevant; (4) Atr demonstrates a similar stiffness range in vitro. We hypothesize that after axon injury, the interaction between the axon tip and the environment

produces a mechanical force that falls into the range capable of activating Piezo, which demonstrates an injury-induced enrichment at the axon tip. This will in turn elicit the downstream signaling to inhibit axon regrowth. In addition, the stiffness of the substrates is determined by the extracellular matrix (ECM) and characteristics of the interacting tissues. Structural ECM molecules, like laminins, collagens, and chondroitin sulfate proteoglycans (CSPGs), are found in the nervous system in mammals and flies[79,80]. Apparently, the stiffness of the ECM is related to the components of those structural ECM molecules and their densities. The tissue environment has also been reported to affect stiffness, and cellular constituents and density may result in a specific stiffness of a certain area. Piezo is identified as a mechanosensor that detects and responds to the environmental stiffness[59], adding another potential layer of regulation during neuroregeneration. Our study does not rule out the possibility that Piezo and Atr may respond to substrate stiffness indirectly. For example, it is potentially possible that the substrate stiffness may affect the axonal cytoskeleton, which in turn regulates Piezo localization, activation, or downstream signaling.

While researchers have successfully regenerated mammalian nerves by targeting intrinsic or extrinsic barriers, these exciting advancements have not yet been translated into successful therapeutics for human patients. This is partly due to our incomplete understanding of the pathways controlling regeneration. Another contributing factor is our limited capabilities to induce proper axon regeneration beyond simply promoting axon regrowth, in order to achieve adequate synapse regeneration and functional recovery. We have thus established a behavioral paradigm in flies, based on the touch sensation, for assessing functional regeneration after CNS injury. It is encouraging to find that inhibiting the Atr-Chek1 pathway presented beneficial effects towards synapse regeneration and functional recovery. It is important to point out that the enhanced behavioral recovery that we observed may directly result from the bona fide regeneration of the injured axons, although it is possible that the sprouting of uninjured axons or circuit level plasticity may also contribute. Importantly, though, the fly sensory neuron injury model offers an ideal opportunity to screen for regeneration regulators, dissect the underlying genetic, cellular, and molecular mechanisms, and test their functional relevance.

Finally, our results showing that pharmacological inhibition of Atr or Chek1 enhanced axon regeneration in cultured mammalian neurons and in flies in vivo provide a strong rationale to further test the viability of this pathway as a potential target for treating neural injury. Importantly, the fact that these pharmacological inhibitors for Atr and Chek1, which are already in clinical trials for cancer therapy, showed efficacy in promoting axon regeneration raises the possibility of repurposing these drugs for regeneration therapy. It is worth mentioning that the inhibitor-induced axon regeneration in vitro that we observed was modest, which may be due, in part, to the simplified environment in the culture dish. In the absence of non-neuronal cell types such as glial cells, which the growth cones interact with, the Piezo-mediated suppression of regeneration is less prominent compared to in vivo.

## Methods

**Fly stocks**. 19-12-Gal4[81], repo-Gal80[82], mei41[29D19], UAS- Cdk1.T14A.Y15F[21], ppk-CD4-tdGFP[83], ppk-Gal4[83], Cdk1[B47 22,23], Cdk1[E1-23 22,23], mei41::FLAG (PBac {fTRG01361.sfGFP-TVPTBF}VK00002)[24], 221-Gal4[84], UAS-His2Av RNAi[31], RPA2[KG00759 32], RPA3[G0241 33], DmPiezoKO, UAS-Piezo RNAiv2796[85], nompC-QF[86], QUAS-mCD8GFP[87], QUAS-mtdTomato[87], ChAT-Gal4[40], nompC-Gal4[86], UAS-synaptotagmin-eGFP[41], Nos[Δ15 88], UAS-mPiezo1-TriM[17], and UAS-Nos[89] have been previously described. grp[A196] P{neoFRT}40 A, UAS-grp RNAiBL27277, UAS-twe, UAS-stg, UAS-twe RNAiBL33044, UAS-twe RNAiBL36587, UAS-mus304 RNAiBL61355, tefu[atm-6], tefu[atm-3], rad50[EP1], nbs[1], mus101[A], UAS-Claspin RNAiBL32974 Hus1-like[MI11259] and ctrip[MI14762], elav-Gal4; UAS-Dcr2, were from the Bloomington stock center. UAS-mei41 RNAi, UAS-grp RNAiv10076, UAS-Rpa-70 RNAiv11210, UAS-Rpa-70 RNAiv110368, UAS-Rpa2 RNAiv102306, UAS-Rpa2 RNAiv30570, UAS-Rpa3 RNAiv101833, UAS-Rpa3 RNAiv15380, UAS-mus304 RNAiv46012, UAS-Rad17 RNAiv103552, UAS-Rad17 RNAiv44723, UAS-Rad1 RNAiv103430, UAS-Rad1 RNAiv12676, UAS-mus101 RNAiv31431, and UAS-Claspin RNAiv34476 were from VDRC. UAS-RpA-70 was from FlyORF. To generate the UAS-hATR-WT, UAS-hATR-KD, UAS-hCHEK1, UAS-RPA2, and UAS-RPA3 stocks, the entire coding sequences were cloned into the pACU2 vector, and the constructs were then injected (Rainbow Transgenic Flies, Inc). Randomly selected male and female larvae were used. Analyses were not performed blind to the conditions of the experiments. In our study, we typically used one mutant plus one RNAi knockdown, or two independent RNAi strains to confirm the phenotype of each candidate gene. In addition, overexpression analysis was performed for critical genes. In order to determine the efficiency of the RNAis, we have measured their knockdown levels using the pan-neuronal elav-Gal4. We collected fly heads or the entire larval CNS tissue and performed quantitative RT-PCR (q-PCR). This strategy allows us to circumvent the issue of low numbers of class III and class IV da neurons in each larva. We found that most of the RNAis show 30–90% knockdown efficiency (Supplementary Fig. 10a). Moreover, to demonstrate that knocking down the DDR genes pan-neuronally is comparable with class III or class IV da neuron-specific knockdown, we show that mus304 RNAi driven by elav-Gal4 also enhances class III da neuron axon regeneration, whereas twe RNAi driven by elav-Gal4 also impedes class IV da neuron axon regeneration (Supplementary Fig. 10b-g).

**Mice**. Atr[fl/fl90] mice were generously provided by E.J. Brown (University of Pennsylvania). Advillin-CreER[91] and Piezo1[fl/fl 92] mice were obtained from Jackson Laboratories. C57BL/6J mice were obtained from the Jackson Laboratory. All studies and procedures involving animal subjects were performed under the approval of the Institutional Animal Care and Use Committee (IACUC) at the Children's Hospital of Philadelphia and Temple University. Four to six-week-old Avil-CreER/Atr[+/+], Avil-CreER/Atr[fl/+], and Avil-CreER/Atr[fl/fl], or Avil-CreER/Piezo1[+/+], Avil-CreER/Piezo1[fl/+], and Avil-CreER/Piezo1[fl/fl] mice were administered 2 mg of tamoxifen daily by intraperitoneal injection for 5 consecutive days. SNL or DRG dissection was performed ~2 weeks after the last tamoxifen injection. Genomic DNA from the dorsal root ganglia was extracted at the end of an experiment and then analyzed by PCR to confirm the deletion. Age-matched mice were randomly assigned to experimental groups. Analyses were not performed blind to the conditions of the experiments. All mice were housed in a vivarium and maintained at consistent temperature (70–74 °F) and humidity (35–55%) on a 12:12 h light-dark cycle. Up to 5 mice of the same sex from the same litter were housed in a cage. The animals had no prior history of drug administration, surgery or behavioral testing.

**Sensory axon lesion in Drosophila**. Da neuron axon lesion and imaging in the PNS or within the VNC were performed in live fly larvae according to published methods[11,12,93]. In brief, larvae were mounted in 90% glycerol under coverslips sealed with grease, and the axon of a da neuron in the periphery or the axon bundle of da neurons in the VNC was targeted using a focused 930 nm two-photon laser (~350–700 mW). Following lesion, animals were recovered to grape juice agar plates and imaged live at the appropriate stages.

**Quantitative analyses of sensory axon regeneration in flies**. Quantification was performed according to published methods[11,12]. Briefly, for axon regeneration in the PNS, we used "Regeneration percentage", which depicts the percent of regenerating axons among all the axons that were lesioned; "Regeneration length", which measures the increase of axon length; "Regeneration index", which is calculated as an increase of "axon length"/"distance between the cell body and the axon converging point (DCAC)" (Supplementary Fig. 1a). The extent of axon regeneration (or retraction) is shown in the schematic (Supplementary Fig 1b). An axon is defined as regenerating only when it obviously regenerated beyond the retracted axon stem, and this was independently assessed of the other parameters. The regeneration parameters from various genotypes were compared to that of the WT if not noted otherwise, and only those with a significant difference were labeled with the asterisks.

**Live imaging in flies**. Live imaging was performed according to published methods[94,95]. Embryos were collected for 2–24 h on yeasted grape juice agar plates and were aged at 25 °C or room temperature. At the appropriate time, a single lava was mounted in 90% glycerol under coverslips sealed with grease, imaged using a Leica SP8 with LAS X or Zeiss LSM880 with Zen 2012, and returned to grape juice agar plates between imaging sessions.

**Behavioral assay**. We used nompC-Gal4/+; nompC-QF, QUAS-CD8GFP/+ larvae as control; nompC-Gal4/UAS-Rad17 RNAiv103552; nompC-QF, QUAS-CD8GFP/UAS-Dcr2 for testing Rad17 knockdown, mus101[A]; nompC-Gal4/+; nompC-QF, QUAS-CD8GFP/+ for testing mus101 mutants, and mei41[29D]; nompC-Gal4/+; nompC-QF, QUAS-CD8GFP/+ for testing mei41 mutants. Larvae were raised at 25 °C and 70% humidity. At 72 h after egg laying (h AEL), larvae were injured at the sites shown in Fig. 4a. After injury, larvae were kept on grape agar plates at 25 °C until analysis.

The behavioral assay for the specific segments was modified from the method described previously[38]. Briefly, an eyelash was used for delivering the gentle-touch stimulus. Based on the contact area between the eyelash and larval body wall, the gentle-touch stimulus was subcategorized into touch+, touch++, and touch+++ (Supplementary Fig. 6a). For each larva, the injured segments (T1 and T2) were stimulated by the eyelash with touch+, touch++, and touch+++, respectively (4 trials for each stimulus), followed by testing of the uninjured segments (T3 and A1). The scores were recorded as "1", if stop (hesitate); "2", if recoil or turn; "3", if one step back (single reverse contractile wave); "4", if two or more steps back (multiple waves of reverse contraction); "0", if no response (Supplementary Fig. 6b). Only those larvae that showed an impaired response when stimulated at the T1 and T2 segments at 8 h AI, but retained normal response at the T3 and A1 segments, were scored. If an injured larva scored >0 in at least two of the four trials, it was defined as showing functional recovery. WT larvae showed no recovery at 24 h AI and only limited recovery at 48 h AI.

**Immunohistochemistry**. Third instar larvae or cultured neurons were fixed according to standard protocols. The following antibodies were used: mouse anti-p-His2Av antiserum (UNC93-5.2.1, 0.3 μg/mL, Developmental Studies Hybridoma Bank), mouse Anti-FLAG M2 (F3165, 1:500, Sigma), rabbit anti-twe (1:100, O'Farrell Lab), mouse anti-Tuj1 (801202, 1:5000, BioLegend), rabbit anti-ATR (13934, 1:500, Cell Signaling), mouse anti-Nucleolin (ab136649, 1:1000, Abcam), rabbit anti-Atrip (PA1-519, 1:400, ThermoFisher), rabbit anti-Chek1 (AV32589, 1:100, Sigma), rabbit anti-TopBP1 (LS-C663420-20, 1:1000, LSBio), α-bungarotoxin, Alexa-594 conjugate (B13423, 5 μg/mL, ThermoFisher), rabbit anti-Gap43 (NB300-143, 1:1000, Novus Biologicals), rabbit anti-SCG10 (NBP1-49461, 1:5000, Novus Biologicals), DAPI (D9542, 1:1000, Sigma), and fluorescence-conjugated secondary antibodies (1:1000, Jackson ImmunoResearch).

**Q-PCR and genotyping of Hus1-like insertion**. Q-PCR was done for Hus1-like and rp49 according to the manufacturer's protocols. To verify the knockdown efficiency of RNAi, elav-Gal4; UAS-Dcr2 females were crossed to each RNAi respectively and the heads of their adult offspring were subjected to q-PCR. The reduction of Rpa-70, Rpa2, or Rpa3 is lethal at the late pupal stage so larval brains were dissected for RNA extraction. For each sample, 30–40 heads/brains were used, with three biological replicates for each genotype. The primers are shown in Supplementary Table 1.

**Cell culture, transfection, treatment, and quantifications**. WT or PIEZO1KO (5E3)[43] HEK293T cells were maintained in DMEM medium supplemented with 10% FBS. Cells were transfected with pcDNA3-FLAG-ATR and GFP (pLL3.7) in the presence of Lipofectamine 2000 (Invitrogen). Forty-eight hours after transfection, cells were exposed to mock or hypotonic medium (medium diluted 1:5 with ddH2O) for 5–7 min. In some conditions, 1400 W dihydrochloride (100 μM, Santa Cruz) as a NOS inhibitor, or histamine (100 μM, Santa Cruz) as a NOS activator, were added to the hypotonic medium. For immunocytochemistry, fixed

with 4% PFA for 20 min, and stained with FLAG antibody (1:500). For fluorescence intensity quantification, the integrated intensity of the whole 8-bit image was measured with ImageJ 1.52q, and the FLAG-ATR fluorescence intensity was normalized to that of GFP. For FLAG-ATR cluster analysis, DAPI was used to define the nuclear area, and the total area of FLAG-ATR clusters per cell was measured using the Analyze Particles plugin (ImageJ). To test endogenous ATR localization before and after hypotonic stress, untransfected WT and *PIEZO1KO* HEK293T cells were treated with hypotonic medium for 15 min before fixed with PFA. Cells were stained with ATR antibody along with the Nucleolin antibody to highlight nucleoli.

**NO imaging**. NO production was detected by DAF-FM diacetate (4-amino-5-methylamino-2′,7′-difluorofluorescein diacetate) (D23844, ThermoFisher). Fly larvae were dissected at 24 or 48 h AI in PBS to expose the sensory neurons. The larval body wall—fillet prep was incubated in 10 µM DAF-FM diacetate for 10 min at room temperature, rinsed in PBS, fixed with 4% paraformaldehyde, rinsed in PBS, and imaged by confocal microscopy. DAF-FM diacetate fluorescence was quantified in ImageJ by measuring the mean gray value around the injured axon tip subtracting the background fluorescence.

**Microfluidic neuron culture and axotomy**. These assays were performed according to published methods[96]. Briefly, a microfluidic culture chamber with several compartments was made using soft lithography in PDMS. The axon compartment and somal compartment were separated by micrometer-sized grooves. Dissociated DRG neurons from E16 rats were plated into the somal compartment. After 7 DIV (days in vitro), neurons in the somal compartment extend axons through the microgrooves and reach the axon compartment. Axotomy was performed by aspiration of the axon compartment. Atr or Chek1 inhibitors were then added to the culture medium and neurons were cultured for another 5 or 18 h within the device, after which cells were fixed and stained with the indicated antibodies.

Quantitative analysis of axon regrowth was performed as described previously[17]. In brief, axons in the terminal chamber labeled by Gap43 staining (1:1000) were imaged using a Zeiss LSM880 laser scanning microscope with a ×40 objective. Axon coverage area was calculated by connecting the tips of the distal axons. Axon coverage area was normalized to the length of the microgrooves to obtain the normalized regeneration.

**Fly larvae injection**. MK-8776 (S2735, Selleckchem, 5 mM stock in DMSO) was dissolved in PBS before use. It was injected into larvae right after injury to a final concentration of ~0.3 µM. The final concentration of the chemical injected into larvae was calculated based on the volume estimated from larvae weight. Injection of PBS with the corresponding DMSO was the vehicle control. Injection was performed by glass micropipettes and Hamilton syringes.

**Sciatic nerve lesion (SNL) and quantification of sensory axon regeneration**. Mice (6–8 weeks old) were anesthetized and a small incision was made on the skin at the mid-thigh level. The sciatic nerve with its three major branches was exposed through a gluteal muscle splitting incision. In the sham control mouse, the posterior tibial nerve was exposed and elevated from its tissue bed, but no crush lesion was performed. In the experimental groups, the nerve crush lesion was performed on the posterior tibial nerve by freeing the nerve from connective tissue and fully crushed for 10 s. The muscle layer was closed with 5-0 chromic gut sutures (Ethicon Inc., NJ) and the skin was closed with Autoclip® system (F.S.T. Instruments, CA).

Three days after surgery, mice were deeply anesthetized with 200 mg/kg ketamine given intraperitoneally followed by exsanguination by trans-cardiac perfusion with PBS (0.01 M, pH 7.4) and fixation by paraformaldehyde (4% PFA in PBS, pH 7.4). Whole DRG (lumbar L3–L5) and full-length sciatic nerve were dissected and post-fixed for 24 h and cryoprotected in sucrose solution (30% sucrose in 0.01 M PBS, pH 7.4) at 4 °C. Fixed DRG and sciatic nerve tissues were embedded in optimal cutting temperature O.C.T. compound (Leica, Germany) and sectioned at 14 µm using Leica CM3050 (Leica, Germany). DRGs were later processed as floating sections and sagittal sciatic nerve sections were thaw-mounted on gelatin-dodecahydrate treated glass microscope slides (Superfrost Plus, Fisher Scientific).

To measure regeneration of the sciatic nerve, sections were stained with the α-SCG10 antibody (1:5000). SCG10 intensity was measured by ImageJ and the average intensities were calculated across 100 µm non-overlapping regions and normalized. The regenerative index was calculated as previously described[97].

**AAV construction and packaging**. We generated the human CHEK1-T2A sequence and inserted it into pscAAV-eGFP. T2A is a "self-cleaving" peptide which enables pscAAV-CHEK1-T2A-eGFP to express CHEK1 and GFP proteins separately. Cloning procedures were performed in the NEB Stable cell line and verified by Srf1 digest. Endotoxin-free pscAAV-CHEK1-T2A-eGFP was packaged into AAV2 by the Research Vector Core at Children's Hospital of Philadelphia. The AAV2 titer is $1.88 \times 10^{13}$ VP/ml.

**Microinjection of AAV vectors into DRGs and sciatic nerve crush**. Recombinant self-complementary AAV2 vectors for eGFP (control) or CHEK1-T2A-eGFP were injected into DRGs at L4 and L5 in C57BL/6 mice using a micropipette connected to a nanoinjector (World Precision Instruments). After the animal was placed in a stereotaxic holder with spinal cord clamps (STS-A, Narishige Group, Japan), the DRGs at L4 and L5 were exposed following laminectomy. A micropipette was introduced 0.5 mm into the DRG and 1.2 µL of AAV2 vector was injected into each DRG over a 10 min period (100 nL/min). The micropipette was left within the injected DRG for 3 min and withdrawn slowly. The wound was closed with 5-0 sterile silk sutures and wound clips. Two weeks after AAV2 injection, the sciatic nerve on the same side of DRG injections was crushed with No. 5 fine forceps for 10 s adjacent to the sciatic notch. The crush site was marked with charcoal-coated forceps and the wound was closed. Animals were perfused for histology 3 days after the sciatic nerve crush.

**Stiffness assay**. Polyacrylamide (PAA) hydrogels were made as previously described[58,98,99] with some modifications. Briefly, 18 mm circular glass coverslips (NeuVitro, Vancouver, WA) were treated with 3-aminopropyl-trimethoxysilane (Fisher Scientific), extensively washed with Milli-Q $H_2O$, and treated with 0.5% glutaraldehyde (Fisher Scientific). Rectangular $22 \times 50$ mm glass coverslips (Fisher Scientific) were treated with hydrocarbon soluble siliconizing solution to make nonadhesive top coverslips. Next, 40% Acrylamide and 2% bis-acrylamide (Bio-Rad Laboratories, Richmond, CA) were mixed at ratios of 5%, 0.1% and 12%, 0.4% to achieve gels of 1.0 and 30.0 kPa stiffnesses, respectively[98]. Polymerization was initiated by adding ammonium persulfate (APS, stock concentration of 10% w/v, ThermoFisher) and N,N,N,N-tetramethylelthylenediamine (TEMED, AcrosOrganics, Morris Plains, NJ). After initiation, 80 µL of gel solution was transferred to each siliconized (bottom) coverslip and covered with the nonadhesive top coverslip. After 15 min, the bottom coverslips were removed, revealing hydrogels attached to only the top coverslip. Gels were allowed to swell in 1x phosphate-buffered saline (PBS) for 1 h at 37 °C, coated in sulfosuccinimidyl6-(40-azido-20-nitrophenylamino) hexanoate (sulfo-SANPAH, ThermoFisher) and photoactivated with 365 nm UV light for 10 min. PAA gels were gently washed with PBS and coated with 0.1 mg/mL Poly-L-Lysine overnight in a 37 °C, 5% $CO_2$, and humidity-controlled incubator. On the day of cell seeding, the PAA hydrogels were coated in 5 µg/mm² laminin (20 µL of 1 mg/mL laminin in 1000 mL growth medium) for 6 h, 37 °C.

DRG primary culture was performed as previously described[57]. Briefly, anesthetized mice were perfused with chilled, sterile PBS, and DRGs tissues were dissected and washed in cold HBSS. The HBSS was replaced with a digestion solution that contained 1 mg L-cysteine (Sigma-Aldrich), 60U Papain (Worthington Biochemical Corporation), 1 µg DNase I (Fisher Scientific) in 1.5 mL $Ca^{2+}/Mg^{2+}$-free HBSS. Tissues were incubated at 37 °C for 7 min, triturated gently, and incubated/triturated twice more. The DRG tissues were pelleted ($250 \times g$, 3 min, room temperature), resuspended, and further digested (37 °C, 20 min) in an enzyme cocktail containing 15 mg dispase II (Sigma-Aldrich) and 12 mg of collagenase (Worthington Biochemical Corporation) in 3 mL of $Ca^{2+}/Mg^{2+}$-free HBSS. The enzyme solution was inactivated by the addition of a 1:1 ratio of tissue solution to prewarmed Neurobasal A medium (1x Glutamax, B27 supplement, 10% FBS). The cell solution was passed through a 100 µm cell strainer (ThermoFisher), pelleted ($250 \times g$, 3 min, room temperature), and resuspended in culture medium (Neurobasal A, B27, Glutamax, 10% FBS, 100 ng/mL neural growth factor; mouse NGF 7.0 s, Millipore Sigma) to achieve the final cell suspension. DRG neurons were then seeded on each corresponding PAA hydrogel at ~10,000 cells per 250 mm² (18 mm round cover glass, Carolina) growth area to achieve single-cell density. Primary DRG neurons were cultured on PAA gels for 38–40 h before fixed in prewarmed 4% sucrose/4% PFA in 1x PBS for 15 min at room temperature. Cells were permeabilized in 0.2% Triton X-100 in TBS (10 min, room temperature) and blocked in 10% normal donkey serum in PBS (2 h, room temperature) and stained with α-Tuj1 antibody (1:5000).

Neurons were imaged on a Zeiss LSM880 confocal laser scanning microscope and image acquisition occurred within 48 h of mounting each hydrogel. Each image's maximum projection image was used during the analysis of neurite lengths. The total neurite length from each neuron was measured with ImageJ's plugin program, "Simple Neurite Tracer"[100] or NeuronStudio 0.9.92, and normalized to the mean of the Control at 1 kPa.

**Statistics and reproducibility**. No statistical methods were used to pre-determine sample sizes but our sample sizes are similar to those reported in previous publications[11,12], and the statistical analyses were done afterward without interim data analysis. Data distribution was assumed to be normal but this was not formally tested. All data were collected and processed randomly. Each experiment was repeated independently at least three times with similar results. The values of "*N*" (sample size) are provided in the figure legends. Data are expressed as mean in bar graphs for percentage, mean ± SEM in scatter plots, or medium in violin plots. No data points were excluded. Two-tailed unpaired Student's *t*-test was performed for comparison between two groups of samples. One-way ANOVA followed by multiple comparison test was performed for comparisons among three or more groups of samples. Two-way ANOVA followed by multiple comparison test was performed for comparisons between two or more curves. Two-sided Fisher's exact

test was used to compare the percentage. Statistical significance was assigned, *P < 0.05, **P < 0.01, ***P < 0.001. Statistical analyses were performed using GraphPad Prism 8.

**Reporting summary**. Further information on research design is available in the Nature Research Reporting Summary linked to this article.

## Data availability

All data supporting the findings of this study are provided within the paper and its supplementary information. All additional information will be made available upon reasonable request to the corresponding author. Source data are provided with this paper.

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

## Acknowledgements

We thank E.J. Brown for the *Atr^{fl/fl}* mouse; T.T. Su for fly lines; Bloomington Stock Center, VDRC and FlyORF for fly stocks; Addgene and GenScript for plasmids; A. Patapoutian for the *PIEZO1* knockout cell line; N. Yakubovich and P. O'Farrell for the twine antibody; G. Jung and Z. Liang for helping with fly dissection; members of the Song lab for helpful discussions. Y.S. is a recipient of the National Institute of Neurological Disorders and Stroke (NINDS) Pathway to Independence Award. This work was supported by an IDDRC New Program Development Award (CHOP/Penn), an NINDS K99/R00 award (R00NS088211), and an NIH grant (1R01NS107392) to Y.S.

## Author contributions

Experimental design, F.L., L.M., D.L., and Y.S.; methodology, F.L., T.Y.L., L.M., Q.W., D.L., K.P., K.M., Y.H., P.A.J., and Y.S.; data collection and analysis, F.L., T.Y.L., L.M., Q.W., H.N.N., D.L., J.N., J.I.G., C.W., S.W., S.T., J.Q., M.B., P.R., G.M.T., S.L., and Y.S.; writing—original draft, D.L. and Y.S.; writing—review & editing, F.L., T.Y.L., L.M., Q.W., G.M.T., and S.L.; funding acquisition, Y.S.; supervision, G.M.T., S.L., and Y.S.

## Competing interests

The authors declare no competing interests.
