## [Peer Review File · Nature Communications]

Reviewers' Comments:

Reviewer #1:

Remarks to the Author:

GENERAL COMMENTS

The paper describes the following findings: 1. Genetic/pharmacological inhibition of ATR or Chk1 promotes neuron regeneration and improves synapse and behavioral recovery, following CNS injury; 2) ATM, DNA damage or RPA do not influence neuronal regeneration; 3) The process also depends on the Piezo mechanosensitive ion channel and its downstream NO signaling. Overall the data provide evidence that a Piezo-ATR-Chk1 axis promotes neuronal regeneration through a process independent on the canonical ssDNA-RPA sensing process. They hypothesize that ATR responds to mechano-stimuli during axon regeneration resulting from the interactions between the growth cone and the environment.

SPECIFIC COMMENTS

1. Line 57 – 58: "We previously..." -Long and confusing sentence. Can be split into two separate sentences to better introduce the model and its outcome. Especially given that the article is based on this model system.

2. Line 72 onwards – Including Drosophila protein nomenclature already here (DDR protein) could be beneficial.

3. Line 133- Class IV phenotypes should be better described for ATR null mutant flies. Is the ATR catalytic activity required? Over expression of a hATR-KD allele and looking at the Class III regeneration should address this issue.

4. Line 155- Supplementary figure 1C – a zoomed-in Image should be provided for the nuclear FLAG staining.

5. Line 159 – If level and localization of ATR are the same in class III and Class IV neurons, regardless of their ability to regenerate, why over-expression of hATR counteracts regeneration only in Class IV neurons?

6. Line 187 – p-His2Av panels should be better represented. Authors should include positive control for p-His2Av staining and supported with additional quantifications (foci counting or percentage of cells with p-His2Av).

7. Line 208 – Can authors show levels of knock down for the DDR proteins used?

8. Line 297 – Figure 6 should be reproduced with endogenous ATR antibodies, as well as colocalization with some nucleoli markers.

9. Line 360 – Figure 8e –error bars missing.

10. the authors should cite a key paper from Nancy Kleckner showing that ATR has HEAT repeats that behave as an elastic connector, ideal to sense mechanical forces.

CONCLUSIONS

This is a potentially interesting paper. The findings are important and the results solid. However, more work is needed to warrant publication in Nat Comm.

Reviewer #2:

Remarks to the Author:

In this manuscript, Li et al. provide first evidence that the Atr-Chek1 pathway, which is known to be involved in DNA damage response, is also involved in regulating neuronal regeneration. They identify the Piezo-NOS pathway, which the group previously showed to regulate neuronal regeneration, as an upstream regulator of Atr. The authors present a large array of carefully conducted experiments, the results are novel and timely, and the manuscript is well-written.

I have a couple of questions, however, which should be addressed to clarify some key points.

1. The authors show that ATR can respond to DNA damage, mechanical stress, and possibly other stimuli. They have furthermore demonstrated that it is not DNA damage that triggers ATR activity in their regeneration model. However, I don't see any evidence for the presence of mechanical stimuli in this system. While the authors refer to mechanical stimuli or mechanical stress elicited after injury in several places, it is not clear which (chronic) mechanical signal it is that the neurons respond to. This should be worked out better.

2. ATR is found in both class III and IV da neurons, while only class IV da neurons can regenerate their axons. How about DmPiezo, is this present only in class III neurons? This should be shown. If DmPiezo is also present in class IV da neurons, it is very likely not a primary regulator of neuronal regeneration but perhaps merely a modulator of some other primary mechanism (possibly what the authors briefly refer to as 'endogenous regenerative drive' in class IV da neurons in the discussion)?

3. Along these lines, is DmPiezo already expressed during development? If so, shouldn't it inhibit the development of functional neuronal networks? This should be addressed.

4. In the in vitro experiments presented in Fig. 8 i and j, it is currently not convincingly demonstrated that axon regeneration is mediated by the Piezo-Atr axis. First of all, as neurons are cultured straight away, I would not call this an assay of axon regeneration, it's rather assessing axon growth (which, however, may use similar pathways). Importantly, the authors should also perturb the Atr-Chek1 pathway on the different substrates and check whether they find similar growth phenotypes as in the Piezo1 cKOs. As a minor point, data in j should not be normalized.

Minor points:

1. Since the study focuses on sensory neurons, the first paragraph of the introduction referring to the CNS is not required.

2. I would like to know why the authors used a Fisher's exact test in some places, as I'm not sure that this is an appropriate test for the data.

3. The discussion could be a bit more concise.

Reviewer #3:

Remarks to the Author:

The manuscript by Li et al. reports a new conserved signaling pathway, Atr-Chek1-Cdc25-Cdk1, as a negative regulator of axon regeneration. Specifically, by using *Drosophila* sensory neuron injury model, the study discovered that Atr-Chek1 acted to inhibit the Cdc25-Cdk1 signaling and thus impede axon regeneration. The study also identified that many genes in the Atr-associated checkpoint complex were also important regulators of axon regeneration. Lastly, it was shown that Atr-Chek1 acted downstream of the Piezo-Nos signaling to inhibit axon regeneration. Overall the study used two classes of sensory neurons with different regeneration ability to perform extensive genetic studies. The experiments resulted in strong evidence supporting the roles of the identified signaling pathway in regulation of axon regeneration. It is not clear how the new signaling pathway was activated and regulated in different types of neurons after axonal injuries. In addition, the evidence that the endogenous Atr acted to regulate mammalian sensory axon regeneration was relatively weak.

1. In Figure 1, it is not clear why class III and IV neurons have different regeneration ability. It seems not due to the different expression levels of the Atr pathway genes. Is it because of different activation of the pathway after axotomy?

2. In Figure 2, 3, the conclusion that Atr-Chek1 regulate axon regeneration independent of DNA damage seems too strong. It is a bit confusing that overexpression of RPA components had no effect on axon regeneration. Could elevated RPA levels alone activate Atr in fly sensory neurons? In response to DAN single-strand break (SSB), does activation of Atr need any other signals besides RPA sensing? Will SSB in class IV neurons inhibit axon regeneration?

3. In Figure 4, 5, the results showed that interfering Chek1 signaling led to enhanced sensory function and synaptogenesis. Were these results specific to Atr-Chek1 signaling or similar to other axon regeneration promoting approaches?

4. In Figure 7, it was shown that after axotomy the level of NO was significantly elevated in class III neurons, thus activating the inhibitory signaling cascade. How about the levels of NO in class IV neurons that can regenerate?

5. In Figure 8, the results showed that deleting Atr in mouse sensory neurons promoted axon regeneration in vivo. Because mouse sensory neurons regenerate their axons robustly after axotomy, did the protein levels of Atr-Chek1 pathway members change in sensory neurons after peripheral nerve injury? In 8g, h, it seems that sensory axon regeneration distance was relatively short compared with published studies (~3-4 mm in average). Will overexpression of Atr or Chek1 in mouse sensory neurons impair axon regeneration?

In 8i, j, the role of Piezo in regulation of substrate stiffness and axon growth was only performed in vitro. Maybe such evidence was not strong enough to support that Piezo can sensing physical signal to regulate axon regeneration.

Reviewer #1:

GENERAL COMMENTS

The paper describes the following findings: 1. Genetic/pharmacological inhibition of ATR or Chek1 promotes neuron regeneration and improves synapse and behavioral recovery, following CNS injury; 2) ATM, DNA damage or RPA do not influence neuronal regeneration; 3) The process also depends on the Piezo mechanosensitive ion channel and its downstream NO signaling. Overall the data provide evidence that a Piezo-ATR-Chek1 axis promotes neuronal regeneration through a process independent on the canonical ssDNA-RPA sensing process. They hypothesize that ATR responds to mechano-stimuli during axon regeneration resulting from the interactions between the growth cone and the environment.

We thank the Reviewer for supporting our finding and conclusion!

SPECIFIC COMMENTS

1. Line 57 – 58: “We previously...” -Long and confusing sentence. Can be split into two separate sentences to better introduce the model and its outcome. Especially given that the article is based on this model system.

We have modified it accordingly.

2.Line 72 onwards – Including Drosophila protein nomenclature already here (DDR protein) could be beneficial.

Thank you for the suggestion. We carefully thought about this and feel that it might be a bit too early to include the fly nomenclature here. Because we are only introducing the Atr-dependent DNA damage pathway in general and have not mentioned fly genotypes/phenotypes. We are of course happy to make the change if the Reviewer feels strongly about it.

3. Line 133- Class IV phenotypes should be better described for ATR null mutant flies. Is the ATR catalytic activity required? Over expression of a hATR-KD allele and looking at the Class III regeneration should address this issue.

We appreciate the Reviewer's suggestion. In our paradigm, we utilized the different regeneration abilities of class IV (able to regenerate) and class III (unable to regenerate) da neurons to identify factors that are required for or inhibit axon regeneration.

Regarding the comment “Class IV phenotypes should be better described for ATR null mutant flies.” Because class IV da neurons normally regenerate robustly, it is not ideal for screening factors that promote regeneration. In other words, it is not ideal to look for phenotypes in a desensitized background. Indeed, we have examined the class IV da neuron regeneration in *Atr/mei41* mutants, but the phenotype is not statistically significant, due to their high regeneration potential to start with. Along the same line, testing inhibitors in class III da neurons, which fail to regenerate, is not informative. This is again because it is impossible to demonstrate inhibition in a neuron that already does not show regeneration. Therefore, our rationale was to test loss-of-function of *Atr/mei41* in class III da neurons, which showed significant promotion of regeneration (Fig. 1a-c). Meanwhile, we tested gain-of-function of *Atr/mei41* in class IV da neurons and found significant reduction of their regeneration (Fig. 1d-f).

Regarding the comment “Is the ATR catalytic activity required? Over expression of a hATR-KD allele and looking at the Class III regeneration should address this issue.” As we explained above, overexpression of hATR-KD in Class III da neurons as the Reviewer suggested will not be informative, because class III da neurons are unable to regenerate, making them unsuited for detecting regeneration inhibition. To address whether the ATR catalytic activity is required, we have overexpressed hATR-WT or hATR-KD in class IV da neurons and found that only the former was able to inhibit regeneration (Fig. 1d-f), indicating that the ATR catalytic activity is necessary for its inhibition on axon regeneration.

4. Line 155- Supplementary figure 1C – a zoomed-in Image should be provided for the nuclear FLAG staining.

We have included zoomed-in images of the nuclear FLAG staining (Supplementary Fig. 1c). Indeed, we were hoping to detect changes of ATR/mei41 localization in the nucleus after axotomy, but did not observe anything obvious.

5. Line 159 – If level and localization of ATR are the same in class III and Class IV neurons, regardless of their ability to regenerate, why over-expression of hATR counteracts regeneration only in Class IV neurons?

We appreciate the comment. As we addressed to point #3, we did not show data on hATR overexpression in class III da neurons because they are already unable to regenerate. As we showed in the manuscript, the regeneration index for class III da neurons in wild-type (WT) is a negative value, which means that their axons mostly did not regenerate, and instead retracted. This makes detection of further inhibiting axon regeneration in class III axons not practical. To clarify, “over-expression of hATR counteracts regeneration **only** in Class IV neurons”, which the Reviewer was concerned about, is not part of our original conclusion. We have provided a schematic diagram corresponding the regeneration index to the axon regeneration extent, to help clarify this issue (Supplementary Fig. 1b).

6. Line 187 – p-His2Av panels should be better represented. Authors should include positive control for p-His2Av staining and supported with additional quantifications (foci counting or percentage of cells with p-His2Av).

Thank you for the suggestion. We have added the positive control for p-His2Av staining (Supplementary Fig. 3a-c) and provided the quantifications (Fig. 2b).

7. Line 208 – Can authors show levels of knock down for the DDR proteins used?

We are keen to address the Reviewer’s concern, but we would also like to mention that the expression analysis in flies is largely limited by the availability of reagents – very few good antibodies exist. How people in the fly field typically resolve this issue, which we practiced in our manuscript, is to use RNAi + mutation, or two independent RNAis for one target gene. Assessing the knockdown efficiency specifically in class III or class IV da neurons, for which there are only <100 cells per larvae, and without the availability of antibodies, is technically challenging. As an alternative, we have measured the knockdown efficiency of all the RNAis used in our manuscript by pan-neuronal knockdown with the *elav-Gal4*. We collected fly heads or the entire larval CNS tissue and performed quantitative RT-PCR. This strategy

allows us to circumvent the issue of low cell number. We found that most of the RNAis show 30% to 90% knockdown efficiency (Supplementary Fig. 10a). Moreover, to demonstrate that knocking down the DDR genes pan-neuronally is comparable with class III or class IV da neuron specific knockdown, we show that *mus304* RNAi driven by *elav-Gal4* also enhances class III da neuron axon regeneration, whereas *twe* RNAi driven by *elav-Gal4* also impedes class IV da neuron axon regeneration (Supplementary Fig. 10b-g). These results have been added to the Methods on P. 31. We hope the Reviewer would find our approach acceptable.

8. Line 297 – Figure 6 should be reproduced with endogenous ATR antibodies, as well as colocalization with some nucleoli markers.

We appreciate the comment. We have repeated the experiment using an antibody that recognizes endogenous ATR, together with a nucleoli marker, as suggested by the Reviewer. Specifically, we examined the endogenous ATR clustering in WT and *PIEZO1KO* cells after hypotonic stimulation. We found that 15 min hypotonic stress induced significantly more ATR clustering in WT cells compared to *PIEZO1KO*, as quantified by the “number of ATR puncta per cell” and the “percent of ATR puncta colocalized with nucleolin” (Supplementary Fig. 7a-c). This result demonstrates that the ATR clustering phenomenon is consistent between the endogenous and ectopically expressed ATR.

9. Line 360 – Figure 8e –error bars missing.

Figure 8e shows regeneration percentage, which is calculated as the percent of axons in the entire population and thus does not have error bars. The same applies to all the other regeneration percentage data in the manuscript, and we used Fisher's exact test for the statistical analysis.

10. the authors should cite a key paper from Nancy Kleckner showing that ATR has HEAT repeats that behave as an elastic connector, ideal to sense mechanical forces

Thanks for the suggestion and we have cited the reference in the Discussion on P. 21.

Reviewer #2:

In this manuscript, Li et al. provide first evidence that the Atr-Chek1 pathway, which is known to be involved in DNA damage response, is also involved in regulating neuronal regeneration. They identify the Piezo-NOS pathway, which the group previously showed to regulate neuronal regeneration, as an upstream regulator of Atr. The authors present a large array of carefully conducted experiments, the results are novel and timely, and the manuscript is well-written.

We thank the Reviewer for the very encouraging and supportive feedbacks!

I have a couple of questions, however, which should be addressed to clarify some key points.

1. The authors show that ATR can respond to DNA damage, mechanical stress, and possibly other stimuli. They have furthermore demonstrated that it is not DNA damage that triggers ATR activity in their regeneration model. However, I don't see any evidence for the presence

of mechanical stimuli in this system. While the authors refer to mechanical stimuli or mechanical stress elicited after injury in several places, it is not clear which (chronic) mechanical signal it is that the neurons respond to. This should be worked out better.

We appreciate the Reviewer's comment. Accumulating evidence has shown that axons and growth cones generate mechanical forces when they are interacting with the substrates during growth (Betz et al., PNAS, 2011, Athamneh et al., Front. Cell. Neurosci., 2015). During the interaction with neurons, stiffness of the environment is an important aspect that determines the extent of mechanical forces which eventually impact on neurons. Piezo, as an ion channel directly gated by mechanical forces, senses a certain range of substrate stiffness in both neural and non-neural cells. In other words, substrates with stiffness within a certain range can activate Piezo channels (Blumenthal et al. P.N.A.S., 2014; Koser et al., Nat. Neurosci., 2016; Chen et al., Neuron, 2018). In our previous work, we have demonstrated that Piezo is activated in the terminals of axotomized axons in *Drosophila* (Song et al., Neuron, 2019), suggesting that the stiffness of the *Drosophila* neural environment is capable of activating Piezo, and that mechanical forces exist between the axotomized axons and the environment.

In this manuscript, we found that, on a substrate with a close-to-*in-vivo* stiffness (reported stiffness of the nerve system in either rodents or *Drosophila* is < 1000 Pa), neurite outgrowth is indeed mediated by Piezo (Fig. 8k, i). However, Piezo did not function when the substrate stiffness is much higher than the *in-vivo* range (Fig. 8k, i). These results further support our hypothesis that a sustained signal for Piezo channels would come from the interaction between axons and the *in vivo* neural environment, both in *Drosophila* and mice.

As no existing approaches would allow direct and precise detection of mechanical forces *in vivo* in the regenerating axon tips, we thought the evidence we provided in our manuscript, together with our previous publication, has demonstrated one of the best links between Piezo activation and substrate stiffness: 1. Piezo is enriched in the axon tips after injury *in vivo* (Song et al., Neuron, 2019); 2. Piezo is activated and regulates the calcium signaling in axons during regeneration *in vivo* (Song et al., Neuron, 2019); 3. *in vitro*, Piezo can impact on neurite outgrowth when the stiffness of substrate is close to that of the neural environment *in vivo*, but not when it is physiologically irrelevant (Fig. 8k, i); 4. A similar substrate stiffness range also exists for Atr (Fig. 8m), confirming that Atr functions downstream of Piezo in inhibiting axon regeneration. In addition, the stiffness of the substrates is determined by the extracellular matrix (ECM) and characteristics of the interacting tissues. Structural ECM molecules, like laminins, collagens and chondroitin sulfate proteoglycans (CSPGs), are found in the nervous system in mammals and flies (Long & Huttner, Open Biol, 2019; Broadie et al., Dev Neurobiol, 2011). Apparently, the stiffness of the ECM is related to the components of those structural ECM molecules and their densities. The tissue environment has also been reported to affect stiffness, and cellular constituents and density may result in a specific stiffness of a certain area. Piezo is identified as a mechanosensor that detects and responds to the environmental stiffness (Koser et al., Nat Neurosci, 2016), adding another potential layer of regulation during neuroregeneration.

We hope this would help address the Reviewer's question. We have added this to the Discussion on P. 23.

2. ATR is found in both class III and IV da neurons, while only class IV da neurons can regenerate their axons. How about DmPiezo, is this present only in class III neurons? This should be shown. If DmPiezo is also present in class IV da neurons, it is very likely not a primary regulator of neuronal regeneration but perhaps merely a modulator of some other primary mechanism (possibly what the authors briefly refer to as 'endogenous regenerative drive' in class IV da neurons in the discussion)?

The Reviewer has raised an excellent point. We have shown that Piezo is also present in class IV da neurons (Song et al., *Neuron*, 2019, Figure S2G). Dozens of intrinsic factors have been identified to regulate the ability of axon regeneration. Some of them are pro-regenerative while others are inhibitory to regeneration (Mahar et al., *Nat. Rev. Neurosci.*, 2018). In our opinion, whether one neuron can regenerate its axon is determined by the synergistic effect of various factors. This could be why different neurons possess distinct regenerative capabilities. A single factor could still be critical even if it is not the dominant mechanism in a specific neuron type. In this manuscript and our previous publication (Song et al., *Neuron*, 2019), the function of DmPiezo has been thoroughly evaluated in the context of axon regeneration, indicating that it is an important factor comparable to other factors that we identified previously (Song et al., *Nat. Neurosci.*, 2015; Song et al., *Genes Dev.*, 2012).

3. Along these lines, is DmPiezo already expressed during development? If so, shouldn't it inhibit the development of functional neuronal networks? This should be addressed.

Thank you for bringing up this point. Piezo proteins have been shown to be expressed throughout development and play a role in various developmental processes, such as vasculature formation and stem cell proliferation (Li et al., *Nature*, 2015; Ranade et al., *PNAS*, 2014; Li et al., *Nature*, 2018). We have tested and discussed the function of Piezo in neural development in our previous publication. We found that Piezo knockout flies are totally viable and fertile, and that the overall axon projections in the CNS, the gross axon projection of da neurons (class IV and class III) in the PNS and CNS, and the single axon terminals of class III da neurons are all normal (Song et al., *Neuron*, 2019, Figure S2A-S2D). In addition, we did not detect obvious patterning defects in sensory neuron specific Piezo1 knockout mice (Song et al., *Neuron*, 2019, Figure S8C). On the other hand, specific knockdown of Piezo after axotomy leads to enhanced axon regeneration (Song et al., *Neuron*, 2019, Figure 3). Consistent with these observations, we found that Piezo is diffusely expressed in the uninjured axons at low levels in flies, but it becomes enriched in the injured axon tip (Song et al., *Neuron*, 2019, Figure 1H). Similarly, Piezo1 is also expressed at very low levels in sensory neurons in mice (Ranade et al., *Nature*, 2014). These led us to hypothesize that while Piezo may be expressed during development, it is likely expressed at a low level unlikely to inhibit the establishment of neuronal networks, or to induce drastic defects when knocked out. After axotomy, however, its enriched expression at the axon tip would allow it to play a more significant role.

Moreover, as we discussed in comment #2, whether a neuron can regenerate its axon is likely not determined by a single factor. Although we can promote or inhibit axon regeneration by manipulating a specific factor, it does not mean that we can predict whether the axon would grow or not by only looking at whether it is expressed.

4. In the *in vitro* experiments presented in Fig. 8 i and j, it is currently not convincingly demonstrated that axon regeneration is mediated by the Piezo-Atr axis. First of all, as neurons are cultured straight away, I would not call this an assay of axon regeneration, it's rather assessing axon growth (which, however, may use similar pathways). Importantly, the authors should also perturb the Atr-Chek1 pathway on the different substrates and check whether they find similar growth phenotypes as in the Piezo1 cKOs. As a minor point, data in j should not be normalized.

Thank you for the suggestion. We agree with the Reviewer that our stiffness assay is a more direct measurement of axon outgrowth *in vitro*, and our rationale is indeed that axon outgrowth and regrowth may use similar pathways. A minor point to add is that during the tissue dissection for culturing, the neuronal processes are removed, which may bear some resemblance to injury. As the Reviewer suggested, we have tested Atr conditional knockout (*Atr cKO*) neurons and found that they show a similar response on the stiffness assay, as the *Piezo1 cKO* (Fig. 8m). Specifically, while Piezo1 is activated at 0.3 to 1 kPa, Atr is more active at 0.3 kPa, suggesting that Atr is likely only one of the signaling branches downstream of Piezo1.

Regarding the data normalization for the stiffness assay, we appreciate the comment. The reason we performed normalization is due to the variability of the stiffness assay across different experiments. In each experiment, we performed the assay by culturing neurons on a same set of substrates with varying stiffnesses. Although the variation among different experiments exists, the interrelationship among different stiffnesses within each experiment is very consistent. Therefore, we used 1 kPa as the internal control and normalization point for each experiment, which allows us to be able to more readily compare the results from multiple experiments.

Minor points:

1. Since the study focuses on sensory neurons, the first paragraph of the introduction referring to the CNS is not required.

We appreciate the suggestion. Our study looked into the regeneration potential of sensory neuron axons both in the PNS and CNS (regarding synapse regeneration and behavioral recovery). This is our rationale to include the introduction about axon regeneration in the CNS.

2. I would like to know why the authors used a Fisher's exact test in some places, as I'm not sure that this is an appropriate test for the data.

We used Fisher's exact test for all the regeneration percentage data. Since the regeneration percentage can be expressed in a way as two variables (numbers of regenerated and non-regenerated), Fisher's exact test is appropriate for the comparison between control and experimental groups. This method has been used in all of our previous publications on regeneration (Song et al., *Genes Dev*, 2012; Song et al., *Nat. Neurosci.*, 2015; Song et al., *Neuron*, 2019; Monahan Vargas et al., *Genes Dev*, 2020).

3. The discussion could be a bit more concise.

We have tightened the Discussion as suggested.

Reviewer #3:

The manuscript by Li et al. reports a new conserved signaling pathway, Atr-Chek1-Cdc25-Cdk1, as a negative regulator of axon regeneration. Specifically, by using *Drosophila* sensory neuron injury model, the study discovered that Atr-Chek1 acted to inhibit the Cdc25-Cdk1 signaling and thus impede axon regeneration. The study also identified that many genes in the Atr-associated checkpoint complex were also important regulators of axon regeneration. Lastly, it was shown that Atr-Chek1 acted downstream of the Piezo-Nos signaling to inhibit axon regeneration. Overall the study used two classes of sensory neurons with different regeneration ability to perform extensive genetic studies. The experiments resulted in strong evidence supporting the roles of the identified signaling pathway in regulation of axon regeneration. It is not clear how the new signaling pathway was activated and regulated in different types of neurons after axonal injuries. In addition, the evidence that the endogenous Atr acted to regulate mammalian sensory axon regeneration was relatively weak.

We thank the Reviewer for the encouraging and constructive comments!

1. In Figure 1, it is not clear why class III and IV neurons have different regeneration ability. It seems not due to the different expression levels of the Atr pathway genes. Is it because of different activation of the pathway after axotomy?

We appreciate the Reviewer's intriguing question on what underlies the differential regeneration ability of class III and class IV da neurons. Investigating the determinant for axon regeneration cell-type specificity is a main direction in the lab, which is a bit beyond the scope of this manuscript. Together with our collaborators, we have published a preprint manuscript focusing on an interesting observation of different responses of class III and class IV da neurons after axotomy, which may shed light on the mechanism ("Patterned Neuronal Activities Dictate Cell Type-specific Axon Regeneration", Ruppell et al., *BioRxiv*, 2019). However, the underlying machinery still remains largely unknown.

As suggested by the Reviewer, we do not think that the differential regeneration ability of class III and class IV da neurons is solely due to the different activation of the Atr pathway, based on the following reasons: 1. inhibiting the Atr pathway in class III da neurons does not enable them to regenerate as robustly as class IV da neurons (Fig. 1a-c), suggesting that the Atr pathway partially contributes to the inability to regenerate in class III da neurons; 2. overexpression of components of the Atr-Chek1 pathway in class IV da neurons does not inhibit their regeneration ability to a similar level of class III da neurons (Fig. 1d-f) – they can still regenerate to some extent with overactivation of the Atr pathway, suggesting that the Atr pathway is not the determinant of the regeneration ability in class IV da neurons; 3. the expression level of Atr/mei41 in class III and class IV da neurons appears to be comparable (Supplementary Fig. 1c). Therefore, we thought our conclusion that "we identified the Atr-Chek1 pathway as a neuronal-intrinsic negative regulator of axon regeneration" in this manuscript is appropriate.

A lot of intrinsic factors have been identified to regulate the ability of axon regeneration, and some of them are pro-regenerative while others are inhibitory to regeneration (Mahar et al., Nat. Rev. Neurosci., 2018). In our opinion, whether one neuron can regenerate its axon is determined by the synergistic effect by various factors. Furthermore, there may be a tug-of-war in neurons after injury, between factors that inhibit regeneration such as the Piezo-Atr pathway, and those that promote regeneration. In class III da neurons, because of their limited intrinsic regenerative potential, the activation of Piezo-Atr is sufficient to suppress further regeneration. However, in class IV da neurons, the regenerative drive is high enough to override this blockade. The pro-regenerative machinery may be determined by differentially expressed calcium channels among various neuronal subtypes (Ruppell et al., BioRxiv, 2019), whereas the Piezo-Atr mediated inhibition may be a more general mechanism in neurons. We have added this to the Discussion on P. 19 and 20.

2. In Figure 2, 3, the conclusion that Atr-Chek1 regulate axon regeneration independent of DNA damage seems too strong. It is a bit confusing that overexpression of RPA components had no effect on axon regeneration. Could elevated RPA levels alone activate Atr in fly sensory neurons? In response to DAN single-strand break (SSB), does activation of Atr need any other signals besides RPA sensing? Will SSB in class IV neurons inhibit axon regeneration?

We agree with the Reviewer and we have softened this conclusion on P. 4, 8 and 9.

For the experiments regarding overexpressing RPA components, we considered the result as a supporting evidence for the conclusion that Atr-Chek1 inhibits axon regeneration independent of DNA damage. Overexpression of RPA would not directly alter ATR activation (Toledo et al. Cell. 2013). But when DNA is damaged, it stimulates the binding of the Rad17 complex to ssDNA, which is required for DNA damage-induced ATR activation (Zou et al. PNAS. 2003). We inferred that, if DNA damage occurs after axotomy, overexpression of RPA would lead to stronger ATR activation and inhibition of axon regeneration. However, our result showed that overexpression of RPA had no effect on axon regeneration (Supplementary Fig. 3d, e), corroborating with our data that loss-of-function of RPAs does not enhance axon regeneration (Fig. 2c-e), and the p-His2Av staining results (Fig. 2a, b and Supplementary Fig. 3a-c) suggesting that axotomy and regeneration do not introduce DNA damage at least in da sensory neurons.

Testing if SSB in class IV da neurons would inhibit axon regeneration is an interesting idea. However, inducing SSB in cells, especially cells in the G0 phase like neurons, and specifically in class IV da neurons, is quite challenging. To our knowledge, there is currently no technology available for this purpose.

3. In Figure 4, 5, the results showed that interfering Chek1 signaling led to enhanced sensory function and synaptogenesis. Were these results specific to Atr-Chek1 signaling or similar to other axon regeneration promoting approaches?

Promotion of functional recovery is not specific to the Atr-Chek1 pathway that we demonstrated here. We have unpublished data indicating other approaches that promote axon regeneration also lead to enhanced recovery of sensory function. We believe that the Atr-Chek1 pathway regulates axon regeneration in a general manner, which is consistent with

our observations that activation or inhibition of the pathway decreases or increases the axon regeneration ability in various types of neurons in fly and mouse.

4. In Figure 7, it was shown that after axotomy the level of NO was significantly elevated in class III neurons, thus activating the inhibitory signaling cascade. How about the levels of NO in class IV neurons that can regenerate?

As suggested, we have performed NO imaging in injured class IV da neurons. We found that NO signal is also present in class IV da neurons after injury, but to a lesser extent – the percent of NO⁺ neurons is lower than that of class III da neurons (Supplementary Fig. 7b, c). This result suggests that while the Piezo-NO-Atr pathway is active in different types of neurons, its strength, which contributes to the regeneration inhibition ability of this pathway, may vary. This is consistent with the different regeneration ability between Class III and Class IV da neurons. We have added this to the Results and Discussion on P. 15 and 20.

5. In Figure 8, the results showed that deleting Atr in mouse sensory neurons promoted axon regeneration in vivo. Because mouse sensory neurons regenerate their axons robustly after axotomy, did the protein levels of Atr-Chek1 pathway members change in sensory neurons after peripheral nerve injury? In 8g, h, it seems that sensory axon regeneration distance was relatively short compared with published studies (~3-4 mm in average). Will overexpression of Atr or Chek1 in mouse sensory neurons impair axon regeneration?

As suggested, we have added expression analysis for Atr and quantified the expression levels of Atr, Atrip, Chek1 and TopBP1 in DRG neurons with and without peripheral nerve injury. We found that the expression of Atr and TopBP1 is modestly increased after sciatic nerve lesion (Supplementary Fig. 5b).

We appreciate the Reviewer's comment regarding the axon regeneration distance. We would like to mention that the regeneration length appears to vary in some recent publications (Pinto-Costa et al., JCI, 2020; Lai et al., Exp Neurol, 2020). In particular, the average regeneration length at the same time point in the Pinto-Costa et al., is ~700 μm, which is comparable to what we observed. We believe that the location of the injury site and the intensity of the crush may contribute to the length difference. To address this concern, we have repeated the sciatic nerve lesion experiment in both WT and *Atr* cKO and found very consistent results. We have now included more representative images (Fig. 8g, h).

Regarding the question whether overexpression of Atr or Chek1 in mouse sensory neurons impairs axon regeneration, we have tested Chek1. We found that human CHEK1 overexpression in DRG neurons (via AAV2) indeed reduces axon regeneration after sciatic nerve lesion (Fig. 8i, j). We noticed that the base line is slightly different between the Atr LOF and CHEK1 overexpression experiments, which may be due to the genetic background and the manipulations, such as tamoxifen injection and virus infection.

In 8i, j, the role of Piezo in regulation of substrate stiffness and axon growth was only performed in vitro. Maybe such evidence was not strong enough to support that Piezo can sensing physical signal to regulate axon regeneration.

We appreciate the Reviewer's comment. First, we would like to mention that, in our previous work, we have demonstrated that Piezo is activated in terminals of axotomized axons in

Drosophila (Song et al., Neuron, 2019), suggesting that the stiffness of *Drosophila* neural environment is capable of activating Piezo, and that mechanical forces exist between the axotomized axons and the environment. In this manuscript, we worked out the function of the Atr pathway in axon regeneration, and interestingly we demonstrated the link between the Atr pathway and the previously identified Piezo in the context of axon regeneration. Second, to our best knowledge, no existing approaches would allow direct detection of mechanical forces *in vivo* in the axon tips. Moreover, data of Figure 8I and 8J (now Fig. 8k-m) is not the only evidence. The description below is the rationale how we approached the question and came to the conclusion:

Accumulating evidence has shown that axons and growth cones generate mechanical forces when they are interacting with the substrates during growth (Betz et al., PNAS, 2011, Athamneh et al., Front. Cell. Neurosci., 2015). During the interaction with neurons, stiffness of the environment is an important aspect that determines the extent of mechanical forces which eventually impact on neurons. Piezo, as an ion channel directly gated by mechanical forces, senses a certain range of substrate stiffness in both neural and non-neural cells. In other words, substrates with stiffness within a certain range can activate Piezo channels (Blumenthal et al. P.N.A.S., 2014; Koser et al., Nat. Neurosci., 2016; Chen et al., Neuron, 2018).

In this manuscript, we found that, on a substrate with a close-to-*in-vivo* stiffness (reported stiffness of the nerve system in either rodents or *Drosophila* is < 1000 Pa), neurite outgrowth is indeed mediated by Piezo (Fig. 8k, i). However, Piezo did not function when the substrate stiffness is much higher than the *in-vivo* range (Fig. 8k, i). These results further support our hypothesis that a sustained signal for Piezo channels would come from the interaction between axons and the *in vivo* neural environment, both in *Drosophila* and mice.

As no existing approaches would allow direct and precise detection of mechanical forces *in vivo* in the regenerating axon tips, we thought the evidence that we provided in our manuscript, together with our previous publication, has demonstrated one of the best links between Piezo activation and substrate stiffness: 1. Piezo is enriched in the axon tips after injury *in vivo* (Song et al., Neuron, 2019); 2. Piezo is activated and regulates the calcium signaling in axons during regeneration *in vivo* (Song et al., Neuron, 2019); 3. *in vitro*, Piezo can impact neurite outgrowth when the stiffness of substrate is close to that of the neural environment *in vivo*, but not when it is physiologically irrelevant (Fig. 8k, i).

To substantiate our conclusion, we have tested additional stiffness points for the *in vitro* experiment (0.16, 0.3, 1, 5 and 30 Kpa), in order to provide a more comprehensive analysis for the range of mechanical forces underlying Piezo activation. Furthermore, we have also tested Atr conditional knockout (*Atr cKO*) neurons and found that they show a similar response on the stiffness assay, as the *Piezo1 cKO* (Fig. 8m). Specifically, while Piezo1 is activated at 0.3 to 1 Kpa, Atr is more active at 0.3 Kpa, suggesting that Atr is likely one of the signaling branches downstream of Piezo1. These have been added to the Results and Discussion on P. 18 and 23.

Reviewers' Comments:

Reviewer #1:

Remarks to the Author:

The authors have addressed my comments/criticisms.

Reviewer #2:

Remarks to the Author:

The authors have addressed all reviewer comments, and the manuscript is very strong now.

A few open points remain, however, which I would like to ask the authors to address in the discussion:

1) The authors state that Piezo is activated by forces, that growth cone forces are higher on stiffer substrates, and yet that Piezo is only activated on soft substrates and not on the stiff ones where forces are higher. This appears to me like a contradiction, I'm not sure if I can follow the argument. This should be discussed.

2) It is still not 100% clear to me how the in vivo results relate to the in vitro data. Are the authors suggesting that tissue stiffness changes after injury?

3) The authors furthermore state 'As no existing approaches would allow direct and precise detection of mechanical forces in vivo in the regenerating axon tips, we believe that the evidence that we have provided, together with our previous publication, has demonstrated one of the best links between Piezo activation and substrate stiffness'. I would also recommend toning this statement down, as (1) such methods exist (e.g. FRET-based force sensors), and (2) other studies have previously demonstrated clear links between Piezo activation and substrate stiffness.

4) I would also suggest to tone down the following statement: 'In the absence of non-neuronal cell types, such as glial cells, which the growth cones interact with, the MECHANICAL FORCE INDUCED SUPPRESSION of regeneration is less prominent compared to in vivo.', as there is currently no direct experimental support of a role of cellular forces in regeneration.

5) It should be clearly stated that DmPiezo is expressed in both class III and IV da neurons, and that it therefore cannot be the primary cause of the differences in regeneration but rather a downstream effector of a currently unknown mechanism.

Reviewer #3:

Remarks to the Author:

The manuscript has been well revised by addressing most of the comments. For the last comments regarding the potential roles of mechanical signal, although the explanation in the revised manuscript was extensive, there are still other possibilities that Piezo and Atr respond to substrate stiffness indirectly. For instance, the substrate stiffness could affect the axonal cytoskeletal organizations, which in turn regulate Piezo localization, activation, and its downstream signaling events. Thus, some discussion and more cautious conclusion would be more appropriate. Overall, the revised manuscript has been greatly improved and is ready to be published.

Reviewer #1 (Remarks to the Author):

The authors have addressed my comments/criticisms.

We thank the Reviewer for supporting our work!

Reviewer #2 (Remarks to the Author):

The authors have addressed all reviewer comments, and the manuscript is very strong now. A few open points remain, however, which I would like to ask the authors to address in the discussion:

1) The authors state that Piezo is activated by forces, that growth cone forces are higher on stiffer substrates, and yet that Piezo is only activated on soft substrates and not on the stiff ones where forces are higher. This appears to me like a contradiction, I'm not sure if I can follow the argument. This should be discussed.

We apologize for the confusion. Even though we know that the function of Piezo channels is to “sense force”, the mechanism of Piezo activation is still under investigation. Recent studies showed that the mechanism is far more complicated than a mere correlation between force and the extent of Piezo activation (Jiang et al., *Trends in Biochemical Sciences*, 2021). Our results in this manuscript and our previous paper (Song et al., *Neuron*, 2019) demonstrate the role of Piezo activation in one of its many facets – axon regeneration. Our results show that Piezo is activated by a range of stiffnesses. The associated conclusion in the manuscript is, based on our *in vitro* neurite outgrowth experiment, Piezo is activated at 0.3 and 1 kPa, but not 5 or 30 kPa. Therefore, there is no increase of neurite outgrowth in *Piezo1 cKO* on substrates of 5 or 30 kPa. This is stated on P. 22-23. Thanks for raising this point, and we have carefully checked the use of words “stiffness” and “force” in the manuscript to avoid the confusion.

2) It is still not 100% clear to me how the *in vivo* results relate to the *in vitro* data. Are the authors suggesting that tissue stiffness changes after injury?

Thanks for the question. We hypothesize that after axon injury, the interaction between the axon tip and the environment produces a mechanical force that falls into the range capable of activating Piezo, which demonstrates an injury-induced enrichment at the axon tip. This will in turn elicit the downstream signaling to inhibit axon regrowth. This is added to P. 23. Tissue stiffness has been reported to change after injury (Moeendarbary et al., *Nature Communications*, 2017), but this is not directly related to our observations or conclusion.

3) The authors furthermore state ‘As no existing approaches would allow direct and precise detection of mechanical forces *in vivo* in the regenerating axon tips, we believe that the evidence that we have provided, together with our previous publication, has demonstrated one of the best links between Piezo activation and substrate stiffness’. I would also recommend toning this statement down, as (1) such methods exist (e.g. FRET-based force sensors), and (2) other studies have previously demonstrated clear links between Piezo activation and substrate stiffness.

Thanks for the suggestion. We have modified the statement accordingly on P. 23 as the following: “As no existing approaches have been demonstrated to have enough resolution of directly and precisely detecting the mechanical forces *in vivo* in the regenerating **axon tips**, we believe that the evidence that we have provided, together with our previous publication, has demonstrated an important link between Piezo activation and substrate stiffness.”

4) I would also suggest to tone down the following statement: ‘In the absence of non-neuronal cell types, such as glial cells, which the growth cones interact with, the MECHANICAL FORCE INDUCED SUPPRESSION of regeneration is less prominent compared to *in vivo*.’, as there is currently no direct experimental support of a role of cellular forces in regeneration.

Thanks for the suggestion. We have modified the statement accordingly on P. 24 as the following: “In the absence of non-neuronal cell types such as glial cells, which the growth cones interact with, the Piezo mediated suppression of regeneration is less prominent compared to *in vivo*.”

5) It should be clearly stated that DmPiezo is expressed in both class III and IV da neurons, and that it therefore cannot be the primary cause of the differences in regeneration but rather a downstream effector of a currently unknown mechanism.

Thanks for the suggestion. We have added the statement in the Discussion on P. 20.

Reviewer #3 (Remarks to the Author):

The manuscript has been well revised by addressing most of the comments. For the last comments regarding the potential roles of mechanical signal, although the explanation in the revised manuscript was extensive, there are still other possibilities that Piezo and Atr respond to substrate stiffness indirectly. For instance, the substrate stiffness could affect the axonal cytoskeletal organizations, which in turn regulate Piezo localization, activation, and its downstream signaling events. Thus, some discussion and more cautious conclusion would be more appropriate. Overall, the revised manuscript has been greatly improved and is ready to be published.

Thanks for the suggestion. We have added the following discussion on P. 23: “Our study does not rule out the possibility that Piezo and Atr may respond to substrate stiffness indirectly. For example, it is potentially possible that the substrate stiffness may affect the axonal cytoskeleton, which in turn regulates Piezo localization, activation, or downstream signaling.”